# S²-ETR: Semantic-Structure Aware Encrypted Traffic Classification via Hyper-Bipartite Graph

## Abstract

In the era of pervasive encryption, encrypted traffic classification serves as a fundamental technique, underpinning diverse applications including intrusion detection and network management. It is commonly approached with deep learning methods that rely on semantic feature extraction or on traffic interaction graphs; however, these approaches suffer from three major limitations: semantic signal obfuscation under strong encryption, which suppresses distinguishable single-flow semantics and undermines accuracy and robustness; inter-flow over-squashing, which constrains the expressivity of interaction graphs and degrades classification performance; and the absence of intra–inter fusion combined with limited scalability, which prevents effective reconciliation of semantic and structural cues and hinders deployment on massive traffic graphs. To address these challenges, we propose S²-ETR (Semantic-Structure Encrypted Traffic Representation), a novel framework that integrates traffic semantics with communication topology graph. The framework includes a Hyper-Bipartite Graph (HBG), which takes two branches to fuse topology and semantic features. The topology branch models structural relations with an IP–flow bipartite graph, decoupling flows from communication entities to mitigate overfitting. The semantic branch employs a lightweight adapter to capture flow semantics, enhancing cross-domain robustness; meanwhile, it constructs semantic hyperedges via implicit hypergraph learning, propagating global semantic representations without extra information. Finally, a conditional probability–based hierarchical classification strategy is introduced to augment scalability on massive traffic graphs. Furthermore, through a mathematical proof, we demonstrate that HBG reduces long-range dependencies and over-squashing, leading to better efficacy and generalization compared to traditional topology graphs. Experimental results show that S²-ETR consistently achieves state-of-the-art performance across 5 datasets of varying scales, outperforming 15 baselines by 2.4%–17.1% on encrypted application classification datasets, and surpassing the best baseline by 9.2% on the more complex and challenging IoT dataset.

## 1 Introduction

In modern network environments, the topology has become increasingly complex, traffic semantics have grown more diverse, and encryption mechanisms are continuously strengthened. As encryption becomes the norm for network communication, the effective payload of packets is often invisible, creating significant challenges for network management, anomaly detection, and security monitoring. Consequently, **Encrypted Traffic Classification (ETC)** has emerged as a crucial task for ensuring Quality of Service (QoS) and empowering early threat detection.

Machine learning–based ETC relies on handcrafted statistical features (Taylor et al. (2016); Van Ede et al. (2020)), which worked under weaker encryption but degrade substantially today. To address this, recent work turns to deep learning. Deep learning–based ETC falls into three families by input representation: (1) Sequence-based methods, which encode packet or byte streams and model temporal dependencies (Wang et al. (2017b); Zhou et al. (2021); Wang et al. (2024)); (2) Image-based methods, which convert traffic into grayscale images and apply image recognition architectures

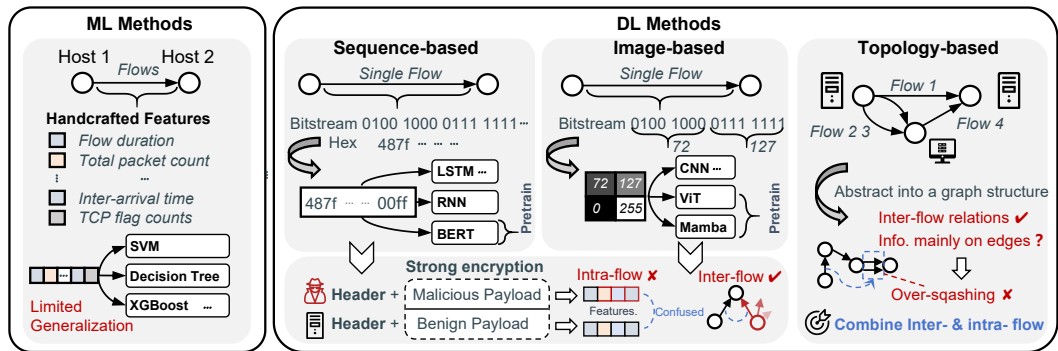

Figure 1: Existing methods and their limitations. Traditional sequence- and image-based approaches focus on intra-flow features and struggle with highly encrypted, complex traffic; topology-based methods capture inter-flow relations but suffer from severe over-squashing. This work proposes a classification method that unifies intra-flow and inter-flow information to overcome these limitations.

(Zhang et al. (2023)); and (3) Topology-based methods, which construct interaction/flow graphs and exploit Graph Neural Networks (GNNs) to capture structural patterns (Zhang et al. (2025)). Pretraining builds on these models by leveraging large-scale traffic corpora to acquire transferable knowledge, further boosting classification under strong encryption (Lin et al. (2022); Zhao et al. (2023)).

Existing methods in encrypted traffic classification (ETC) have made significant progress, but they still face several challenges, as shown in Fig. 1. Traditional machine learning methods rely on handcrafted statistical features, which are difficult to tune and often fail to generalize to unseen encryption schemes and application behaviors. Sequence-based and image-based deep models are designed to capture the semantics of individual flow byte sequences. However, recent studies (Fu et al. (2023)) indicate that some carefully crafted malicious flows become almost indistinguishable from benign flows after encryption. Their byte sequences are highly randomized and exhibit high entropy, which makes them difficult to differentiate at the single-flow level. These observations motivate a shift in ETC from modeling isolated flows to exploiting relations among flows, commonly through topology-based analyses. In traffic interaction graphs, most discriminative information resides on edges rather than nodes. During message passing, information must traverse intermediate nodes, which aggravates over-squashing and scalability issues, especially when multiple flows between the same IP pair are collapsed into a single edge embedding. Consequently, existing topology-based GNN methods have seen limited adoption, particularly for highly encrypted traffic and large-scale datasets that demand more efficient processing. Therefore, we aim to jointly model intra-flow semantics and inter-flow relations to achieve more accurate detection and better generalization across diverse encrypted traffic scenarios.

To address these limitations, we propose a general-purpose framework for ETC, named **Semantic-Structure Encrypted Traffic Representation (S²-ETR)**, which does not rely on external prior knowledge. S²-ETR consists of three main components: a flexible **adapter** for encoding traffic semantics, a **Hyper-Bipartite Graph (HBG)** that integrates semantic hyperedges with a decoupled IP–flow bipartite structure, and a **classifier** module to accommodate datasets of diverse scales. The HBG is structured with two complementary branches, semantic and topological, which jointly capture both the semantic and structural characteristics of traffic flows, as shown in Fig. 2. In the semantic branch, adapter-encoded flow features are refined via Implicit Hypergraph Learning (IHL), which propagates semantics across IP pairs and induces relations among flows with similar patterns. In the topological branch, each flow is modeled as an independent edge between its corresponding IP nodes, mitigating over-squashing and redundant aggregation. By integrating the semantic and topological branches and building upon IHL, S²-ETR improves stability and predictive accuracy, and can efficiently adapt to datasets of varying scales without handcrafted priors or costly pretraining. For ultra-large-scale networks, S²-ETR further maintains competitive performance through a hierarchical classifier guided by conditional probabilities.

**Contributions.** We summarize them as follows:

- We propose $S^2$-ETR, a general-purpose framework for encrypted traffic representation that enables abstract and universal flow-level representations. Without relying on handcrafted features or external prior knowledge, it iteratively updates flow semantics via IHL, attaining robust efficacy across diverse traffic scenarios.

- We design a **Hyper-Bipartite Graph (HBG)** that integrates trainable semantic hyperedges with a decoupled IP–Flow bipartite structure, jointly catching both semantic and topological information. Furthermore, we provide **a mathematical proof** that *HBG reduces long-range dependencies and over-squashing*, offering better performance and generalization than traditional topology graphs.

- We conduct comprehensive experiments on 5 datasets, demonstrating that $S^2$-ETR attains higher accuracy and efficiency compared with 15 state-of-the-art approaches. Furthermore, it maintains strong performance on ultra-large-scale networks by employing a hierarchical classifier supervised with conditional probabilities.

## 2 RELATED WORKS

Recent work on encrypted traffic classification (ETC) has increasingly focused on learning semantic representations that reflect network behavior patterns implicitly encoded (Xie et al. (2023)) in observable byte streams, packet sequences, and flow statistics (Shen et al. (2022)). Early machine learning (ML)–based approaches rely on handcrafted statistical features, such as APPScanner (Taylor et al. (2016)) and Flowprint (Van Ede et al. (2020)). Although handcrafted features were effective, they are sensitive to parameter settings and often fail to generalize to unseen encryption schemes and application behaviors. To alleviate manual feature engineering, recent deep learning (DL) methods transform traffic into alternative representations and apply specialized architectures for classification. Sequence-based models, including ET-BERT (Lin et al. (2022)) and TrafficFormer (Zhou et al. (2024)), encode raw bytes or packet sequences as tokens to learn semantic patterns on individual flows. Image-based methods map flows to two-dimensional or grayscale images and exploit image recognition backbones to capture spatial structures (Hang et al. (2023); Zhao et al. (2023); Shapira & Shavitt (2021)). However, recent studies indicate that, after encryption, some carefully crafted malicious flows become almost indistinguishable from benign flows; their byte sequences are highly randomized and exhibit high entropy, which severely limits the discriminative power of single-flow semantics (Fu et al. (2023)). These observations motivate shifting the focus from isolated semantics to relations among flows.

Topology-based models thus construct interaction graphs between hosts and flows and employ GNNs to leverage the communication structure (Zhou et al. (2021); Zheng et al. (2022); Okonkwo et al. (2025)). In encrypted traffic interaction graphs, most discriminative information resides on edges (i.e., flows), while nodes mainly serve as connectors. Representative models such as E-GraphSAGE (Lo et al. (2022)) and ST-Graph (Fu et al. (2022)) assign weights or importance scores to edges and then aggregate them via intermediate host nodes, which forces information exchange between two flows to always pass through at least one host and inevitably discards fine-grained edge attributes. As the number of layers increases, long-range dependencies are compressed into narrow node embeddings, leading to pronounced over-squashing. Existing remedies are mainly based on rewiring, such as the curvature-based approach SDRF (Topping et al. (2021)) and the community-based method ComFy (Rubio-Madrigal et al. (2025)), which can alleviate over-squashing by improving connectivity, but they are not tailored to large, edge-centric encrypted traffic graphs and may introduce nontrivial computational overhead. We further discuss and empirically compare our approach with rewiring-based methods in the experimental section.

## 3 METHOD

### 3.1 FRAMEWORK OF THE PROPOSED $S^2$-ETR

As illustrated in Fig. 3, our framework consists of three components: (a) *Flow Representation via a Hyper-Bipartite Graph*, (b) *Flow Semantic Awareness via Adapter*, and (c) *Fusion of Topology and Semantic Branches*.

**(a) Flow Representation via a Hyper-Bipartite Graph.**

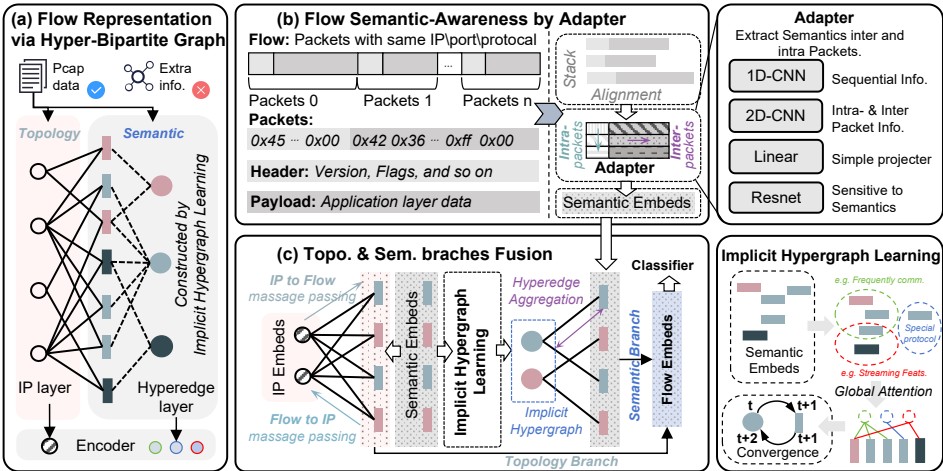

Figure 2: A **Hyper-Bipartite Graph (HBG)** as the foundational graph structure for $S^2$-ETR, abstracting network flows into a graph-based formulation.

Figure 3: Framework of the proposed $S^2$-ETR, which integrates IP-flow bipartite structure with Semantic-Structure implicit hypergraph learning.

Without any external information, HBG decouple IP nodes and flow nodes and connect them to form an IP–flow bipartite graph, as shown in Fig. 2. This structure links each IP to its associated flows and avoids the information loss of traditional topology-based methods that compress multiple edges between the same pair of hosts. Flow-node embeddings are first extracted from raw byte streams and then aggregated via a hyperedge attention mechanism.

**(b) Flow Semantic Awareness via Adapter.**

The semantic branch aims to extract features from individual flows. A flow is defined as a sequence of packets sharing the same source IP/port and destination IP/port, and each packet consists of a header and a payload. To capture both temporal and structural patterns, we crop and align packet headers and payloads, then stack them into a two-dimensional sequence. The horizontal axis encodes the temporal order within each packet, while the vertical axis aligns the same byte positions across packets (e.g., the evolution of flag bits). This 2D representation compactly summarizes the characteristics of a single flow. In principle, any architecture capable of processing two-dimensional sequences can serve as the adapter. In our experiments, we instantiate the adapter with several representative backbones: 1D-CNN for sequence modeling, 2D-CNN for spatial modeling, a linear layer as a basic baseline, and ResNet for deeper feature extraction. Details are shown in Appendix. D.1.

**(c) Fusion of Topology and Semantic Branches.**

The topology branch performs message passing on the hyper-bipartite graph, propagating information between IP and flow nodes to capture how hosts interact through flows. The semantic branch adopts Implicit Hypergraph Learning (IHL), a special case of implicit graph learning where node representations are defined as the fixed point of a hypergraph propagation operator (a concise formulation is given in Section 3.2.2). We initialize IHL with the adapter-derived semantics and use an attention mechanism to parameterize hyperedge weights, iteratively updating node states until con-

vergence. Finally, we fuse the topology and semantic branches using a sparsity-aware static weight derived from the local IP–flow ratio, producing the final representations for downstream detection and improving both accuracy and generalization.

## 3.2 HYPER-BIPARTITE GRAPH (HBG)

The Hyper-Bipartite Graph (HBG) is designed to jointly model the topological and semantic structures of traffic flows, powering efficient message passing and mitigating GNN over-squashing.

### 3.2.1 IP-FLOW BIPARTITE GRAPH FOR TOPOLOGY AWARENESS.

The topological component of HBG is constructed as an IP–flow bipartite graph $\mathcal{G}_B = (\mathcal{U}_{\text{ip}}, \mathcal{V}_{\text{flow}}, \mathcal{E}_B)$. Here, $\mathcal{U}_{\text{ip}}$ denotes the set of IP nodes and $\mathcal{V}_{\text{flow}}$ denotes the set of flow nodes. For each flow with a source IP and a destination IP, we introduce a flow node and connect it to its two endpoint IP nodes, forming the corresponding edges in $\mathcal{E}_B$. In this way, each flow is represented as an independent edge between its two IPs, and flows that share the same IP endpoints have the same IP neighborhood but may still differ in ports or protocols, reflecting different stages or types of communication.

The biadjacency matrix $\mathbf{A}_B \in \{0, 1\}^{n_u \times n_v}$ encodes the connections, with $\mathbf{A}_B(p, q) = 1$ if and only if $(u_{\text{ip}}^{(p)}, v_{\text{flow}}^{(q)}) \in \mathcal{E}_B$. Flow embeddings are aggregated from IP embeddings via

$$\mathbf{H}_{\text{ip2flow}} = \mathbf{A}_B^\top \mathbf{H}_{\mathcal{U}_{\text{ip}}} \in \mathbb{R}^{|\mathcal{V}_{\text{flow}}| \times d_B}, \tag{1}$$

where $\mathbf{H}_{\mathcal{U}_{\text{ip}}} \in \mathbb{R}^{|\mathcal{U}_{\text{ip}}| \times d_B}$ contains IP embeddings.

### 3.2.2 IMPLICIT HYPERGRAPH LEARNING VIA ITERATIVE GLOBAL SEMANTIC UPDATING

Hypergraph learning traditionally captures high-order relations among nodes via explicitly predefined hyperedges. Conversely, our approach eliminates the need for an incidence matrix to propagate information. Instead, we implicitly learn these relationships via an iterative attention mechanism, refining flow node and hyperedge embeddings without relying on explicit hyperedge definitions.

Let $\mathcal{V}_{\text{flow}}$ signify the flow node set, and let $\mathcal{E}_{\text{he}}$ designate a hypothetical hyperedge set. Instead of exploiting a fixed node-hyperedge transformation matrix, we define the implicit relationship between flow nodes and hyperedges as a learned process through attention mechanisms.

At each iteration $t$, the embeddings of flow nodes $\mathbf{H}_{\text{flow}}^t$ and hyperedges $\mathbf{H}_{\text{he}}^t$ are updated via an attention-based propagation scheme:

$$\mathbf{H}_{\text{flow}}^{t+1} = \sigma\left(\mathbf{A}_{\text{fh}}^{(t)} \mathbf{H}_{\text{he}}^t\right), \qquad \mathbf{H}_{\text{he}}^{t+1} = \sigma\left(\mathbf{A}_{\text{hf}}^{(t)} \mathbf{H}_{\text{flow}}^t\right), \tag{2}$$

where $\mathbf{A}_{\text{fh}}^{(t)}$ and $\mathbf{A}_{\text{hf}}^{(t)}$ are attention-based propagation matrices between flow nodes and hyperedges, and $\sigma(\cdot)$ denotes a non-linear activation function. The matrices $\mathbf{A}_{\text{fh}}^{(t)}$ and $\mathbf{A}_{\text{hf}}^{(t)}$ are constructed from query and key vectors $\mathbf{Q}$ and $\mathbf{K}$ using a softmax normalization; the exact formulations are provided in Appendix D.2. After convergence of the iterative updates in equation 2, we obtain the refined flow semantic embeddings, denoted as $\mathbf{H}_{\text{flow}}'$.

Choudhuri et al. (2025) proves that the structure of implicit hypergraph neural networks can solve a fixed-point equilibrium equation through iteration for representation learning. With this conclusion, we leverage the capacity of IHL to refine the embeddings of flow nodes and hyperedges for implicitly evolving and improving each other via hypergraph attention. This iterative process of mutual optimization renders the model to learn the relationships among flow nodes in a highly dynamic and context-sensitive manner, without utilizing prior information.

### 3.2.3 FEATURE FUSION: INTEGRATION OF TOPOLOGICAL AND SEMANTIC EMBEDDINGS

Let $\mathbf{H}_{\text{ip2flow}} \in \mathbb{R}^{|\mathcal{V}_{\text{flow}}| \times d_B}$ symbolize the flow embeddings obtained from the topological branch, and $\mathbf{H}_{\text{flow}} \in \mathbb{R}^{|\mathcal{V}_{\text{flow}}| \times d_S}$ denote the flow embeddings from the semantic branch.

To fuse these two sources of information, we define a structure-aware weighting factor

$$\alpha = \frac{|\mathcal{V}_{\text{flow}}|}{|\mathcal{V}_{\text{flow}}| + |\mathcal{U}_{\text{ip}}|} \in [0, 1], \tag{3}$$

where $|\mathcal{V}_{\text{flow}}|$ and $|\mathcal{U}_{\text{ip}}|$ are the numbers of flows and IPs, respectively. Intuitively, $\alpha$ measures the relative density of flows per IP: a larger $\alpha$ indicates that each IP participates in multiple flows, reflecting a more complex network, whereas a smaller $\alpha$ suggests a flatter topology with many IPs involved in only a few flows. In addition to this static, structure-aware weighting, we also compare two representative dynamic fusion mechanisms, namely a Gated Multimodal Unit (GMU)–based method and an Attentional Feature Fusion (AFF)–based method; the corresponding results and discussions are presented in Section 4.4.

The final fused embedding for each flow is then computed as a convex combination:

$$\mathbf{H}_{\text{HBG}} = \alpha \, \mathbf{H}_{\text{ip2flow}} + (1 - \alpha) \, \mathbf{H}'_{\text{flow}} \in \mathbb{R}^{|\mathcal{V}_{\text{flow}}| \times d_F}, \tag{4}$$

where $d_F = \max(d_B, d_S)$ or after linear projection to align dimensions.

This guarantees that in structurally complex networks ($\alpha \uparrow$), the topological embeddings $\mathbf{H}_{\text{ip2flow}}$ dominate the final representation, leveraging rich connectivity patterns. In contrast, in flattened or sparse networks ($\alpha \downarrow$), semantic embeddings $\mathbf{H}'_{\text{flow}}$ become more influential, capturing high-order flow correlations beyond direct topological links.

### 3.3 CLASSIFICATION HEADS AND LOSS DESIGN FOR VARYING-SCALE DATA

**Standard flow classification.** Given the fused flow embeddings $\mathbf{H}_{\text{HBG}} \in \mathbb{R}^{|\mathcal{V}_{\text{flow}}| \times d_F}$, the prediction probabilities can be obtained as $p_{\text{flow}} = \text{softmax}(\mathbf{H}_{\text{HBG}}\mathbf{W} + \mathbf{b}) \in \mathbb{R}^{|\mathcal{V}_{\text{flow}}| \times N}$, where $N$ is the class number, and $\mathbf{W}, \mathbf{b}$ are learnable parameters. We optimize $p_{\text{flow}}$ by standard cross-entropy, which is straightforward and applicable to small- or medium-scale datasets.

**Hierarchical classification for large-scale datasets.** We consider large-scale ETC with hierarchical labels, where the number of coarse classes is denoted by $N_c$ and the number of fine classes by $N_f$, and let $M \in \{0, 1\}^{N_c \times N_f}$ indicate the coarse-to-fine class mapping, where $M_{ij} = 1$ if fine class $j$ belongs to coarse class $i$.

Given fused flow embeddings $\mathbf{H}_{\text{HBG}} \in \mathbb{R}^{|\mathcal{V}_{\text{flow}}| \times d_F}$, we compute coarse- and fine-level logits simultaneously: $\mathbf{z}_c = \mathbf{H}_{\text{HBG}}\mathbf{W}_c + \mathbf{b}_c$, $\mathbf{z}_f = \mathbf{H}_{\text{HBG}}\mathbf{W}_f + \mathbf{b}_f$, and define the marginalized fine-level probability $p_f^{(n)}(j) = \sum_{i=1}^{N_c} \text{softmax}(\mathbf{z}_c^{(n)})_i \, p_{f|c}^{(n)}(j|i)$, where $p_{f|c}^{(n)}(j|i)$ is the conditional fine-level probability given coarse class $i$ (*cf.* Appendix E for full definition). The hierarchical classification loss then combines coarse- and fine-level supervision with optional bilinear regularization:

$$\mathcal{L}_{\text{hier}} = \underbrace{\mathcal{L}_c + \lambda \mathcal{L}_f}_{\text{Cross-entropy on coarse and fine labels}} + \underbrace{\gamma \, \mathcal{R}_{\text{bilinear}}}_{\text{Bilinear consistency}}, \tag{5}$$

where $\lambda, \gamma$ balance the contributions of fine-label supervision and regularization, and detailed forms of $\mathcal{L}_c, \mathcal{L}_f, \mathcal{R}_{\text{bilinear}}$ are given in Appendix E. The S$^2$-ETR algorithm is illustrated in Appendix C.

### 3.4 THEORETICAL ANALYSIS: WHY IS HBG BETTER THAN IP TOPOLOGY GRAPH?

Hyper-Bipartite Graph reduces the need for long-range dependencies present in the original IP Topology Graph, and thereby avoids the over-squashing phenomena in MPNN (Message Passing Neural Network). MPNN and receptive field are widely used in graph learning, whose detailed definition is displayed in Eqs. (12) & (13). By the chain rule, $h_i^{(\ell)} = h_i^{(\ell)}(x_1, \ldots, x_n)$ is differentiable in $X$ if $\phi_\ell, \psi_\ell$ are differentiable. Following Jake Topping et. al. Topping et al. (2022), over-squashing can be assessed via the Jacobian entry $\frac{\partial h_i^{(r)}}{\partial x_s}$: small sensitivity to distant inputs indicates severe information compression along the path.

**Lemma 1.** *(Sensitivity bound Topping et al. (2022)) Assume an MPNN as defined in Eq. (12). Let $i, s \in V$ with $s \in S_{r+1}(i)$. If $\|\nabla \phi_\ell\| \le \alpha$ and $\|\nabla \psi_\ell\| \le \beta$ for $0 \le \ell \le r$, then*

$$\left| \frac{\partial h_i^{(r+1)}}{\partial x_s} \right| \le (\alpha\beta)^{r+1} (\hat{A}^{r+1})_{is}. \tag{6}$$

Detailed proof of Lemma 1 and subsequent derivations are provided in Appendix B.

**Relation between receptive field and over-squashing:** Lemma 1 shows that, under bounded derivatives, message influence depends on powers of $\hat{A}$. As $r$ grows, $(\hat{A}^{r+1})_{is}$ typically decays (e.g., exponentially on trees), causing over-squashing. In Hyper-Bipartite Graph (HBG), a shallow fusion MLP directly integrates $\mathbf{H}_{\text{ip2flow}}$ and $\mathbf{H}_{\text{he2flow}}$, enabling their connection.

**Reduction of long-range dependencies:** For two flows $r$ hops apart in the IP graph, an MPNN requires at least $r$ layers, and Lemma 1 bounds the sensitivity, which vanishes when $r$ is large or crosses bottlenecks. In HBG, the route is constant depth: one hop IP→Flow conveys endpoint IP context to its flow, and one hypergraph attention layer relays Flow→Hyperedge→Flow, coupling related flows independent of $r$. Thus, end-to-end flow dependency is achieved in two aggregation layers, regardless of IP-level distance in $G_{\text{IP}}$.

**Implication for over-squashing:** In our proposed HBG model, the aggregation of IP-flow bipartite graph $\mathbf{H}_{\text{ip2flow}} = \mathbf{A} \cdot \mathbf{H}_{\text{ip}}$ can be regarded as a one-layer MPNN, and the hypergraph attention aggregation layer $\mathbf{H}_{\text{he2flow}} = \text{ATT}(\mathbf{H}_{\text{flow}}, \mathbf{E}_{\text{he}})$ can be regarded as a one-layer MPNN on the weighted dense bipartite graph. Finally, the multi-order fusion layer performs integration for the two MPNN embeddings. The Bipartite Hypergraph formulation replaces potentially long IP-level paths by constant-depth communication via IP→Flow and Flow↔Hyperedge aggregations.

The design of the proposed HBG model reduces the long-range dependencies, because it only needs two one-layer MPNNs and an MLP layer to connect the two types of learned hidden information. In the original IP Topology Graph, if the distance between two nodes is $D_G(i, j) = r$, it requires MPNN with at least $r$ layers to connect them. While in our HBG model, we only need two one-layer MPNNs to connect them.

Consequently, the reliance on long-range message passing is reduced, and the Jacobian-based attenuation associated with over-squashing is avoided. This establishes the claim of the remark.

## 4 EXPERIMENT

### 4.1 EXPERIMENTAL SETTINGS

**Implementation details.** Regarding input representation, the first 6 packets of each flow were selected, with 80 bytes allocated to headers and 240 bytes to payloads as flow features. Models were trained for 120 epochs with early stopping of 20 epoch patience, learning rate $=1 \times 10^{-3}$, batch size = 32, and dropout rate = 0.2. For performance evaluation, we followed common practices in ETC (e.g., filtering samples larger than 1 KB). Since certain baselines demand excessive memory and cannot be executed on large-scale datasets, fairness is ensured by employing the full ISCX-VPN2016 (Gil et al. (2016)) dataset and randomly sampling 500 flows* per class from the

Table 1: **Datasets and tasks.**

| Dataset | Flow Num | IP Nodes | Categories |
|---|---|---|---|
| **Performance comparison** | | | |
| CIC-IoT2023* | 4000 | 1051 | 8 |
| ISCX-VPN2016 | 4538 | 681 | 12 |
| USTC-TFC2016* | 8000 | 5799 | 16 |
| CipherSpectrum2025 | 41025 | 778 | 41 |
| **Scalability** | | | |
| CIC-IoT2023 | 86423 | 3891 | 8 |
| **Large-scale** | | | |
| CIC-AndMal2017 | 663032 | 7914 | 43 |

CIC-IoT2023 (Neto et al. (2023)) and USTC-TFC2016 (Wang et al. (2017c)) sets, as shown in Tab. 1. We further conducted varying scale tests (10%–100%) on CIC-IoT2023, complemented by real-time evaluation, robustness testing, ablation studies. Furthermore, CipherSpectrum2025 (Wickramasinghe et al. (2025)) is adopted to assess model performance under highly encrypted traffic and advanced cipher suites, while CIC-AndMal2017 (Lashkari et al. (2018)) is employed for large-scale encrypted malware traffic experiments.

**Metrics and baselines.** The evaluation metrics include Accuracy (ACC) and Macro F1-score (F1). Comparative analysis is performed against four categories of state-of-the-art baselines: (1) **ML-based methods**. Flowprint (Van Ede et al. (2020)), AppScanner (Taylor et al. (2016)), XGBoost (Chen & Guestrin (2016)), which apply classical machine-learning models to handcrafted statistical features; (2) **Sequence-based methods**. 1D-CNN (Wang et al. (2017b)), TSCRNN (Lin et al. (2021)), Hast (Wang et al. (2017a)), ET-BERT (Lin et al. (2022)), TrafficFormer (Zhou et al. (2024)),

which treat traffic as packet/token sequences and learn temporal or contextual representations; (3) **Image-based methods**. 2D-CNN (Zhou et al. (2021)), FlowPic (Shapira & Shavitt (2019)), YaTC (Zhao et al. (2023)), NetMamba (Wang et al. (2024)), which transform flows into 2D images and apply convolutional architectures; (4) **Topology-based methods** MH-Net (Zhang et al. (2025)), TFE-GNN (Zhang et al. (2023)), E-GraphSAGE (Mirlashari & Rizvi (2024)), which operate on graph-structured traffic to capture structural and topological dependencies.

## 4.2 COMPARISON WITH STATE-OF-THE-ART METHODS

**Quantitative performance analysis.** As presented in Tab. 2, $S^2$-ETR consistently surpasses DL, pretraining, and GNN-based baselines in terms of ACC and F1 across all four benchmarks (CIC-IoT2023, ISCX-VPN2016, USTC-TFC2016, and CipherSpectrum2025). The best $S^2$-ETR variants achieve ACC/F1 of 0.8650/0.8652 on CIC-IoT2023, 0.9537/0.9512 on ISCX-VPN2016, 0.9812/0.9812 on USTC-TFC2016, and 0.9802/0.9802 on CipherSpectrum2025. Compared with the strongest non-$S^2$-ETR baseline YaTC, $S^2$-ETR improves ACC by about 14.3, 4.2, 0.5, and 0.9 percentage points on the four datasets, respectively, with highly consistent F1 gains. Relative to the average DL baselines, the ACC improvement on CIC-IoT2023 exceeds 20 percentage points, illustrating substantial benefits even on challenging IoT traffic. Notably, these gains hold on both high-encryption (CIC-IoT2023 under TLS 1.3) and low-encryption (ISCX-VPN2016 under TLS 1.2) settings, as well as on both sparse (USTC-TFC2016) and dense (CIC-IoT2023) topologies. On the newly proposed CipherSpectrum2025 dataset, $S^2$-ETR (ResNet) pushes ACC/F1 to 0.9802/0.9802, further narrowing the error margin compared with the already strong baselines and confirming the effectiveness of $S^2$-ETR in more realistic encrypted traffic scenarios.

**Model design advantages.** Across datasets in Tab. 2, the four $S^2$-ETR variants show complementary ACC/F1 behaviors that reflect their architectural roles. On CIC-IoT2023, $S^2$-ETR (2D-CNN) achieves the best ACC/F1 (0.8650/0.8652), about 2.4 points higher in ACC than 1D-CNN, indicating the benefit of cross-packet semantic modeling in complex IoT traffic. On ISCX-VPN2016 and USTC-TFC2016, $S^2$-ETR (ResNet) attains the highest ACC/F1 (0.9537/0.9512 and 0.9812/0.9812), suggesting that multi-level semantic fusion better exploits IP–flow relations. The Linear variant, though simplest, still reaches strong ACC/F1 (e.g., 0.9460/0.9329 on ISCX-VPN2016), outperforming most baselines and showing that even direct IP–flow integration is effective. On CipherSpectrum2025, $S^2$-ETR (ResNet) again delivers the highest ACC/F1 (0.9802/0.9802), outperforming both other $S^2$-ETR variants and all baselines. Overall, these ACC/F1 trends confirm that sequential encoding, cross-packet CNNs, and multi-level fusion jointly provide robust gains on diverse encrypted traffic benchmarks.

**Dataset-specific numerical highlights.** As evidenced in Tab. 3, our framework outperforms 15 baselines by 2.4%–17.1% on encrypted application classification datasets, and researches a 9.2% improvement over the best baseline (E-GraphSAGE) on the more complex and challenging IoT dataset. For VPN, our model shows gains of 4.2% ACC / 3.6% F1 over YaTC, and 17.8% ACC / 17.0% F1 over the DL average. On TFC, where the topology is sparse, $S^2$-ETR still leads with a 0.5% ACC / 0.6% F1 improvement over YaTC, revealing that semantic encoding effectively compensates for weaker structural cues.

## 4.3 COMPREHENSIVE EVALUATION

**Scalability.** Fig. 4 and Appendix Tab. 4 illustrate the performance of $S^2$-ETR across varying training data ratios (10%–100%). Even with just 10% of the data, adapters already exhibit strong performance, with ACC ranging from 0.772 to 0.790, demonstrating good efficiency at smaller scales. As the data scale increases, performance improves steadily, with ACC rising to 0.887–0.909 at 100%. The 2D-CNN excels at small scales (e.g., 0.7895 at 10%), catching strong cross-packet semantics, while ResNet and Linear models perform best at larger scales (up to

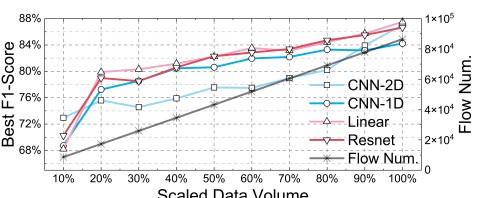

Figure 4: Scalability evaluation results.

Table 2: Model performance comparison on four datasets. The best result is highlighted in **bold**.

| Model | CIC-IoT2023 | | ISCX-VPN2016 | | USTC-TFC2016 | | CipherSpectrum2025 | |
|---|---|---|---|---|---|---|---|---|
| | ACC | F1 | ACC | F1 | ACC | F1 | ACC | F1 |
| Flowprint | 0.6350 | 0.6312 | 0.8038 | 0.8005 | 0.3563 | 0.3091 | 0.1045 | 0.0910 |
| Appsanner | 0.5241 | 0.4097 | 0.7247 | 0.7242 | 0.8302 | 0.6715 | 0.6550 | 0.7671 |
| XGBoost | 0.7114 | 0.5610 | 0.9077 | 0.8888 | 0.9415 | 0.9245 | 0.7856 | 0.7835 |
| 1D-CNN | 0.7000 | 0.6978 | 0.8007 | 0.7771 | 0.9706 | 0.9704 | 0.7268 | 0.7411 |
| 2D-CNN | 0.6725 | 0.6715 | 0.7808 | 0.7533 | 0.9675 | 0.9674 | 0.7918 | 0.7959 |
| FlowPic | 0.6725 | 0.6716 | 0.7930 | 0.7606 | 0.9600 | 0.9599 | 0.7406 | 0.7468 |
| Hast | 0.6500 | 0.6493 | 0.7775 | 0.7587 | 0.9613 | 0.9610 | 0.3009 | 0.3296 |
| TSCRNN | 0.6238 | 0.6205 | 0.7258 | 0.6834 | 0.8669 | 0.8686 | 0.7258 | 0.7347 |
| YaTC | 0.7225 | 0.7210 | 0.9121 | 0.9161 | 0.9762 | 0.9763 | 0.9715 | 0.9714 |
| ET-BERT | 0.6525 | 0.6659 | 0.8656 | 0.8386 | 0.9250 | 0.9352 | 0.9544 | 0.9564 |
| TrafficFormer | 0.6415 | 0.6420 | 0.8082 | 0.7398 | 0.9520 | 0.9533 | 0.9660 | 0.9658 |
| TFE-GNN | 0.4448 | 0.4620 | 0.8796 | 0.8589 | 0.9646 | 0.9624 | 0.8736 | 0.8600 |
| MH-GNN | 0.4808 | 0.4610 | 0.8620 | 0.8302 | 0.9269 | 0.8711 | 0.8905 | 0.8763 |
| E-GraphSAGE | 0.7770 | 0.7729 | 0.7638 | 0.6516 | 0.9110 | 0.9074 | 0.8317 | 0.8036 |
| Netmamba | 0.7299 | 0.7241 | 0.8948 | 0.8938 | 0.9745 | 0.9742 | 0.8340 | 0.8346 |
| $S^2$-ETR (1D-CNN) | 0.8413 | 0.8404 | 0.9405 | 0.9200 | 0.9794 | 0.9793 | 0.9780 | 0.9779 |
| $S^2$-ETR (2D-CNN) | **0.8650** | **0.8652** | 0.9438 | 0.9329 | 0.9506 | 0.9501 | 0.9725 | 0.9723 |
| $S^2$-ETR (Linear) | 0.8413 | 0.8418 | 0.9460 | 0.9329 | 0.9706 | 0.9706 | 0.9708 | 0.9709 |
| $S^2$-ETR (ResNet) | 0.8612 | 0.8609 | **0.9537** | **0.9512** | **0.9812** | **0.9812** | **0.9802** | **0.9802** |

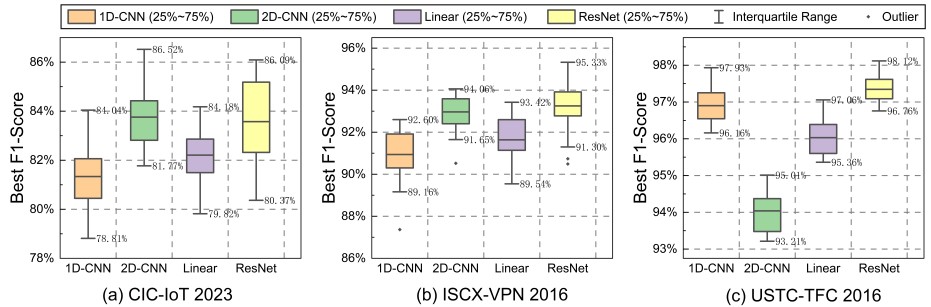

Figure 5: Boxplot of robustness across adapters.

0.9009/0.9089), reflecting the ability to fuse semantics effectively and integrate IP–flow data. 1D-CNN retains stable performance across scales (0.7831→0.8867). In summary, $S^2$-ETR holds better scalability with consistent improvements across multiple model variants, substantiating its potential for processing data of various sizes.

**Robustness.** We assessed this stability across 25 runs and 3 datasets (IoT, VPN, TFC). As illustrated in Fig. 5 and Appendix Tab. 5, 2D-CNN yields the most consistent efficacy on IoT traffic, while ResNet excels on VPN and TFC traffic with top accuracy and low variance. Quantitatively, ResNet has 0.9537/0.9812 ACC on VPN/TFC, and 2D-CNN leads on IoT (0.8650), with deviations $\leq 0.02$ and variances $\leq 3.0\times10^{-4}$. F1 scores follow the same trend, validating $S^2$-ETR robustness in diverse encrypted traffic scenarios.

**Real-time ability and complexity test.** $S^2$-ETR consistently outperforms baselines in speed and memory efficiency (Tab. 6). The Linear adapter is the fastest at 0.0179 ms ($6\times$ faster than HAST) and less than 42 MB memory. $S^2$-ETR (1D-CNN) balances speed and size (0.0282 ms, 34.68 MB), passing FlowPic (0.1056 ms, 74.99 MB) and TSCRNN (0.7267 ms, 15.01 MB), expressing resource-efficient real-time performance. In terms of computational complexity (Tab. 7), $S^2$-ETR variants use more parameters and FLOPs than baselines but remain efficient: $S^2$-ETR (ResNet) has 3.320,M parameters and 89.962,M FLOPs, while the 1D-CNN variant provides a lighter option with 437.834,K parameters and 12.112,M FLOPs. This trade-off between accuracy and computational cost underlines the need for scalable, resource-efficient designs.

**Large-scale study.** On CIC-AndMal2017 (43 categories), hierarchical $S^2$-ETR variants substantially outperform flat baselines (Tabs. 8 & 9). The best variant (hier+1D-CNN) achieves 0.4174/0.4038 ACC/F1 on the 43-class task. On the 5-class task, performance reaches 0.6955/0.6011, exceeding TSCRNN by more than 500%. NetMamba, with 1.859 M parameters and ACC/F1 = 0.4134/0.2580 and 0.4134/0.3120 at 2.653 ms, attains comparable accuracy but remains notably slower than our 1,ms $S^2$-ETR. These results highlight the benefit of explicitly modeling hierarchical structure to jointly capture coarse- and fine-grained semantics in malware classification.

### 4.4 ABLATION STUDY ON SEMANTIC ADAPTER, BRANCH FUSION, AND IHL

**Branch Ablation.** We implemented ablation experiments to inspect the contribution of each module (Tab. 10). **(a) Full model stands for the complete framework,** incorporating the topology and implicit hypergraph learning (IHL) components, which outperforms its ablation variants. **(b) Without topology branch variant**, the performance was markedly reduced, particularly on CIC-IoT2023, where accuracy drops from 0.8650 to 0.4912, highlighting the critical role of IP-level structural priors. **(c) Without semantic branch variant**, it also causes performance degradation (e.g., F1 drops from 0.8652 to 0.8403 on CIC-IoT2023), though the impact is less severe, verifying the complementary role effectiveness of hypergraph modeling.

Fig. 6 visualizes the feature space distribution of our $S^2$-ETR and its three ablation experiments after training on IoT. The t-SNE results prove that $S^2$-ETR effectively distinguishes classes in complex, highly encrypted datasets, whereas the k-nearest neighbor GNN struggles due to its lack of adaptability. In addition, as depicted in Tab. 10, **(d) without IHL variant** (replace the IHL with a *k-nearest neighbor GNN*), it brings about drastic performance degradation (e.g., ACC of 0.4000 on CIC-IoT2023). This variant relies solely on fixed semantic similarity between flow embeddings, forcing semantically close nodes to aggregate even when they belong to different classes, which amplifies noise. In comparison, our **Implicit Hypergraph Learning** adaptively learns and refines the connectivity, powering correct message propagation and generating more discriminative representations.

**Comparison of different over-squashing remedies.** We compare our IHL approach against two representative baselines for mitigating over-squashing: the curvature-based SRDF method and the community-based ComFy method, evaluated on the state-of-the-art CipherSpectrum dataset. All methods share the same underlying topological graph and semantic node initialization; they differ only in the semantic aggregation architecture. As shown in Table 11, our method outperforms ComFy by 3% and SRDF by 0.3%. We attribute these gains to the ability of IHL to more effectively alleviate over-squashing, enabling better long-range information propagation while preserving essential local structure.

**Comparison of Static and Dynamic Fusion.** For the globally defined static fusion coefficient $\alpha$, we compare two dynamic alternatives, GMU and AFF, both of which learn adaptive fusion weights through trainable parameters. As reported in Table 12, the static-weight strategy is both faster and more effective: it achieves approximately 3.5% higher accuracy while being 70%–80% faster. This is because the learnable fusion weights in the dynamic variants tend to converge prematurely and get trapped in suboptimal local minima, whereas the static weight, which is designed to reflect the global topological sparsity, can maintain more stable and superior performance.

## 5 CONCLUSION

We propose a semantic-structure aware framework for encrypted traffic representation, with adaptability to multiple scale traffic data, strong generalization capacity, and no reliance on prior information. Our framework incorporates three key components, including an adaptive Adapter, a Hyper-Bipartite Graph (HBG), and a scale-specific classifier. Empirical results reveal that our method outperforms baselines in different data scales, encryption strength, and application scenario settings, along with strong robustness and real-time efficiency. Our approach offers remarkable advantages for applications such as intrusion detection and quality of service monitoring.

## 6 REPRODUCIBILITY STATEMENT

The code for our model can be accessed through the following Anonymous GitHub link: https://anonymous.4open.science/r/S2-ETR-961E/.

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

## A  LLM USAGE STATEMENT

During the preparation of this work, the authors utilized ChatGPT-4o for proofreading. Following its use, the authors thoroughly reviewed and edited the content as necessary and take full responsibility for the final publication.

## B  SUPPLEMENTARY PROOF FOR THE PROPERTIES OF HBG

### B.1  PRELIMINARIES: BIPARTITE GRAPH AND HYPERGRAPH

**Definition 1** (Bipartite Graph). *Let $\mathcal{G}_B = (\mathcal{U}_B, \mathcal{V}_B, \mathcal{E}_B)$ denote a bipartite graph, where $\mathcal{U}_B$ and $\mathcal{V}_B$ are disjoint node sets, $\mathcal{U}_B \cap \mathcal{V}_B = \emptyset$, and $\mathcal{E}_B \subseteq \mathcal{U}_B \times \mathcal{V}_B$ contains edges connecting nodes across the two partitions. Bipartite graphs are widely used to model interactions between two distinct entity types, and their adjacency can be compactly represented by a sparse matrix $A_B \in \{0, 1\}^{|\mathcal{U}_B| \times |\mathcal{V}_B|}$. If the nodes in $\mathcal{V}_B$ are associated with embeddings $\mathbf{H}_{\mathcal{V}_B} \in \mathbb{R}^{|\mathcal{V}_B| \times d_B}$, where $d_B$ denotes the feature dimension, the embeddings of $\mathcal{U}_B$ can be aggregated through the bipartite structure as*

$$\mathbf{H}_{\mathcal{U}_B} = A_B \mathbf{H}_{\mathcal{V}_B} \in \mathbb{R}^{|\mathcal{U}_B| \times d_B}, \tag{7}$$

*which provides compact and general expressions for message passing across two partitions.*

**Definition 2** (Hypergraph). *Let $\mathcal{G}_{\mathcal{H}} = (\mathcal{V}_{\mathcal{H}}, \mathcal{E}_{\mathcal{H}})$ denote a hypergraph, where $\mathcal{V}_{\mathcal{H}} = \{v_i \mid i \in I_v\}$ is the node set and $\mathcal{E}_{\mathcal{H}} = \{e_i \mid i \in I_e \wedge e_i \subseteq \mathcal{V}_{\mathcal{H}} \wedge e_i \neq \emptyset\}$ is the hyperedge set, where $I_v$ and $I_e$ are the index sets of nodes and hyperedges, respectively. Unlike ordinary graphs where edges connect exactly two nodes, $e_j$ capture higher-order relationships among multiple nodes simultaneously.*

*The connectivity between nodes and hyperedges can be compactly represented by an incidence matrix $H \in \{0, 1\}^{|\mathcal{V}_{\mathcal{H}}| \times |\mathcal{E}_{\mathcal{H}}|}$, where $H_{ij} = 1$ if node $v_i \in \mathcal{V}_{\mathcal{H}}$ belongs to $e_j$, and $H_{ij} = 0$ otherwise. Information propagation on hypergraphs is typically implemented via Hypergraph Neural Networks (HGNN), which can be abstractly expressed as*

$$\mathbf{X}' = \text{HGNN}(\mathcal{G}_{\mathcal{H}}, \mathbf{X}), \tag{8}$$

*where $\mathbf{X} \in \mathbb{R}^{|\mathcal{V}_{\mathcal{H}}| \times d_{\mathcal{H}}}$ denotes the input node embeddings, $d_{\mathcal{H}}$ is the feature dimension, and $\mathbf{X}' \in \mathbb{R}^{|\mathcal{V}_{\mathcal{H}}| \times d_{\mathcal{H}}}$ represents the updated embeddings.*

In our encrypted traffic representation, hyperedges are defined implicitly via a self-supervised mechanism based on semantic similarity between flows. Each hyperedge acts as a "community center," and nodes attend to similar hyperedges using global attention. This allows effective high-order information propagation without relying on external knowledge or additional preprocessing. In our framework, hyperedge embeddings are learned via self-supervised attention on flow semantics, serving as community centers that enable global high-order propagation without extra supervision or preprocessing.

**Definition 3** (Implicit Graph Learning)**.** *Given an input graph $G_0 = (V, E_0)$ with node features $X \in \mathbb{R}^{|V| \times d}$, implicit graph learning refers to a family of methods that do not rely on a fixed, explicitly given adjacency matrix for message passing. Instead, they learn a task-dependent latent graph*

$$A_\theta = \Phi_\theta(X, G_0), \tag{9}$$

*where $\Phi_\theta$ is a parameterized operator (e.g., attention, similarity, or neural kernel) that produces edge weights or connectivity patterns on the fly. The latent graph $A_\theta$ is optimized jointly with the encoder and task head, so that both the graph structure and node representations are adapted end-to-end to the downstream objective.*

Implicit hypergraph learning *extends this idea from pairwise graphs to higher-order relations. Given a (possibly empty) initial hypergraph $H_0 = (V, \mathcal{E}_0)$, the model learns a latent hypergraph*

$$\mathcal{H}_\theta = \Psi_\theta(X, H_0), \tag{10}$$

*where $\Psi_\theta$ produces a (soft) incidence structure or hyperedge weights that encode multi-node interactions without explicitly enumerating all hyperedges in advance. Message passing is then performed on $\mathcal{H}_\theta$, allowing the model to capture higher-order dependencies in an implicit, task-driven manner.*

**Definition 4** (Encrypted Traffic Classification)**.** *Consider the set of network flows $\mathcal{T} = \{t_0, t_1, \ldots, t_n\}$, where each flow $t_i \in \mathcal{T}$ (with $i = 0, \ldots, n$) consists of $m_i$ packets $t_i = \{\mathrm{pkt}_i^0, \mathrm{pkt}_i^1, \ldots, \mathrm{pkt}_i^{m_i}\}$ and is uniquely identified by the 5-tuple ($s_{\mathrm{IP}}$: source IP, $s_{\mathrm{port}}$: source port, $d_{\mathrm{IP}}$: destination IP, $d_{\mathrm{port}}$: destination port, $p$: transport protocol). Each packet $\mathrm{pkt}_i^j$ can be decomposed into a header and a payload. Header information captures protocol version, flags, and other metadata, while the payload contains application-layer data. The header and payload features of flow $t_i$ are represented as matrices $\mathbf{X}_{\mathrm{hdr}}^{(i)} \in \mathbb{R}^{m_i \times d_{\mathrm{hdr}}}$ and $\mathbf{X}_{\mathrm{pld}}^{(i)} \in \mathbb{R}^{m_i \times d_{\mathrm{pld}}}$, where $m_i$ is the packet number in $t_i$, and $d_{\mathrm{hdr}}$ and $d_{\mathrm{pld}}$ denote the feature dimensions of header and payload.*

*Let $\mathcal{Y} = \{y_0, y_1, \ldots, y_k\}$ denote the set of traffic class labels, where each label corresponds to a specific application or attack type. The encrypted traffic classification task is to learn a mapping*

$$f_{\mathrm{cls}} : (\mathbf{X}_{\mathrm{hdr}}^{(i)}, \mathbf{X}_{\mathrm{pld}}^{(i)}) \mapsto y_i, \quad (\mathbf{X}_{\mathrm{hdr}}^{(i)}, \mathbf{X}_{\mathrm{pld}}^{(i)}) \in \mathbb{R}^{m_i \times (d_{\mathrm{hdr}} + d_{\mathrm{pld}})}, \quad i = 0, \ldots, n, \tag{11}$$

*which assigns each $t_i \in \mathcal{T}$ a label $y_i \in \mathcal{Y}$ based on its packet-level header and payload features.*

## B.2 THEORETICAL DERIVATION OF HOW HBG MITIGATES OVER-SQUASHING

**Definition of MPNN:** Let $G = (V, E)$ be a graph endowed with node features $X = (x_1, \ldots, x_n)$, where $x_i \in \mathbb{R}^d$ is the feature at node $i$. For $\ell \geq 0$, denote by $h_i^{(\ell)}$ the hidden representation of node $i$ at layer $\ell$, initialized by $h_i^{(0)} = x_i$. Given differentiable message aggregation function $\psi_\ell$ and update function $\phi_\ell$, a generic MPNN layer Gilmer et al. (2017) is:

$$h_i^{(\ell+1)} = \phi_\ell \left( h_i^{(\ell)}, \sum_{j=1}^{n} A_{ij} \psi_\ell(h_i^{(\ell)}, h_j^{(\ell)}) \right). \tag{12}$$

**Remark 1.** *Bipartite Hypergraph reduces the need for long-range dependencies present in the original IP Topology Graph, and thereby avoids the over-squashing phenomena in MPNN.*

**Definition of receptive field:** Consider a simple, undirected, connected graph $G = (V, E)$, with $(i, j) \in E$ iff $i \sim j$. Let $A$ be the adjacency, $\tilde{A} = A + I$ the self-loop augmented adjacency, $\tilde{D} = D + I$ the augmented degree matrix, and $\hat{A} = \tilde{D}^{-1/2} \tilde{A} \tilde{D}^{-1/2}$ the normalized augmented adjacency (self-loops are standard in MPNNs). For $i \in V$ and $r \in \mathbb{N}$, define

$$S_r(i) := \{ j \in V : d_G(i, j) = r \}, \qquad B_r(i) := \{ j \in V : d_G(i, j) \leq r \}, \tag{13}$$

where $d_G$ is the shortest-path metric. The set $B_r(i)$ is the *receptive field* of an $r$-layer MPNN at node $i$.

By the chain rule, $h_i^{(\ell)} = h_i^{(\ell)}(x_1, \ldots, x_n)$ is differentiable in $X$ if $\phi_\ell, \psi_\ell$ are differentiable. Following Jake Topping et. al. Topping et al. (2022), over-squashing can be assessed via the Jacobian entry $\frac{\partial h_i^{(r)}}{\partial x_s}$: small sensitivity to distant inputs indicates severe information compression along the path.

**Relation between receptive field and over-squashing:** Lemma 1 shows that, under bounded derivatives, message influence is governed by powers of $\hat{A}$. As the required number of layers $r$ grows, the factor $(\hat{A}^{r+1})_{is}$ typically decays (e.g., exponentially on tree-like regions), manifesting over-squashing for long-range dependencies.

**HBG as constant-depth message routing:** In HBG, two structural aggregations are used:

- IP→Flow on the IP–Flow bipartite graph: with biadjacency $\mathbf{A}$, the aggregation

$$\mathbf{H}_{\text{ip2flow}} = \mathbf{A}\,\mathbf{H}_{\text{ip}} \tag{14}$$

  is a one-layer MPNN layer from IPs to flows (each flow collects from its two endpoint IPs).
- Hyperedge→Flow on the Flow–Hyperedge layer: with learnable hyperedges $\mathbf{E}_{\text{he}}$, the attention

$$\mathbf{H}_{\text{he2flow}} = \text{ATT}(\mathbf{H}_{\text{flow}}, \mathbf{E}_{\text{he}}) \tag{15}$$

  can be regarded as one MPNN layer on a weighted dense bipartite structure (flows $\leftrightarrow$ hyperedges), enabling Flow and Hyperedge communication within one hop.

A shallow fusion MLP then integrates $\mathbf{H}_{\text{ip2flow}}$ and $\mathbf{H}_{\text{he2flow}}$, and it enable the connection between $\mathbf{H}_{\text{ip2flow}}$ and $\mathbf{H}_{\text{he2flow}}$.

**Reduction of long-range dependencies:** Consider two flows whose endpoints in the original IP graph are at distance $r$, an MPNN must be at least $r$ layers deep to propagate signals across that path, and Lemma 1 implies a sensitivity bounded by $r$, which can be very small when $r$ is large or when the path crosses bottlenecks.

In contrast, in HBG, the effective route is of constant depth:

1. One hop IP→Flow delivers each endpoint IP's context to its flow in a single layer (no need to traverse $r$ IP hops).
2. One hypergraph attention layer relays information Flow→Hyperedge→Flow among semantically related flows, completing the coupling in constant sub-hops (independent of $r$).

Therefore, the end-to-end dependency between the two flows is realized within two aggregation layers, irrespective of their IP-level distance in $G_{\text{IP}}$.

**Implication for over-squashing:** In our proposed HBG model, the aggregation of IP-flow bipartite graph $\mathbf{H}_{\text{ip2flow}} = \mathbf{A} \cdot \mathbf{H}_{\text{ip}}$ can be regarded as a one-layer MPNN, and the hypergraph attention aggregation layer $\mathbf{H}_{\text{he2flow}} = \text{ATT}(\mathbf{H}_{\text{flow}}, \mathbf{E}_{\text{he}})$ can be regarded as a one-layer MPNN on the weighted dense bipartite graph. Finally, the multi-order fusion layer performs integration for the two MPNN embeddings. The Bipartite Hypergraph formulation replaces potentially long IP-level paths by constant-depth communication via IP→Flow and Flow↔Hyperedge aggregations.

The design of the proposed HBG model reduces the long-range dependencies, because it only needs two one-layer MPNNs and an MLP layer to connect the two types of learned hidden information. In the original IP Topology Graph, if the distance between two nodes is $D_G(i, j) = r$, it requires MPNN with at least $r$ layers to connect them. While in our HBG model, we only need two one-layer MPNNs to connect them.

Consequently, the reliance on long-range message passing is reduced, and the Jacobian-based attenuation associated with over-squashing is avoided. This establishes the claim of the remark.

## C  Algorithm of S$^2$-ETR

## D  Detailed Flow Preprocessing and Encoding

**Input Preprocessing and Anonymization.** Each packet is grouped into bytes in $\mathcal{H} = \{0, \ldots, 255\}$:

$$\mathbf{S}^{(i)} \in \mathcal{H}^{m_i \times l/8}, \quad \mathbf{S}^{(i)\prime} = \mathcal{A}(\mathbf{S}^{(i)}), \tag{16}$$

where $\mathcal{A}(\cdot)$ replaces all source/destination IPs with 255.255.255.255 and ports with 0.

---

**Algorithm 1:** S$^2$–ETR: Semantic-Structure Encrypted Traffic Representation

---

**Input:** Flow $f = (header, payload)$, both represented in hexadecimal format
**Output:** Predicted label $\hat{y}$

**Step 1: Bipartite Graph Initialization**
Extract source $ip_s$, and destination $ip_d$ from $header$;
Initialize Flow nodes $v_f$ and IP nodes $v_{ip_s}, v_{ip_d}$;
Update adjacency $A \leftarrow A \cup \{(v_{ip_s}, v_f), (v_f, v_{ip_d})\}$;

**Step 2: Alignment**
$header \leftarrow \text{Anonymize}(header)$ with $ip = 255.255.255.255, port = 0$;
Serialize flow $s_f \leftarrow \text{HexMap}(header, payload) \in [0, 255]^L$;

**Step 3: Semantic Embedding via Adapter**
Stack packets: $X_f \in \mathbb{R}^{m \times d}$;
Semantic embedding: $h_f \leftarrow \text{Adapter}(X_f)$;

**Step 4: Graph Learning with Implicit Hypergraph Attention**
Initialize a set of $k$ hyperedges $\mathcal{E} = \{e_1, \ldots, e_k\}$ with learnable attributes $\mathbf{H}_e \in \mathbb{R}^{k \times d_h}$.
**for** *each training iteration* **do**

> **Semantic Branch:**
> Compute attention scores $\mathbf{A}_{f,e} = \text{softmax}\big((\mathbf{H}_f \mathbf{W}_Q)(\mathbf{H}_e \mathbf{W}_K)^\top / \sqrt{d_h}\big)$ between flow embeddings $\mathbf{H}_f \in \mathbb{R}^{N_f \times d_h}$ and hyperedge attributes.
> Obtain semantic embeddings $\mathbf{H}'_{\text{flow}} = \mathbf{A}_{f,e} \mathbf{H}_e$.
> Update hyperedge attributes as $\mathbf{H}_e \leftarrow \mathbf{H}_e + \Delta \mathbf{H}_e$, where $\Delta \mathbf{H}_e$ is optimized by back-propagation.
> **Topology Branch:**
> Aggregate embeddings of connected IPs to obtain topology embeddings $\mathbf{H}_{\text{ip2flow}} = \mathbf{A}_B^\top \mathbf{H}_{\mathcal{U}_{\text{ip}}}$, where $\mathbf{A}$ is the bipartite adjacency matrix and $\mathbf{H}_{\mathcal{U}_{\text{ip}}}$ the IP embeddings.
> **Fusion:**
> Fuse the two branches as $\mathbf{H}_{\text{HBG}} = \alpha \mathbf{H}_{\text{ip2flow}} + (1 - \alpha)\mathbf{H}'_{\text{flow}}$, with $\alpha$ a learnable parameter.

**Step 5: Classification**
$\hat{y} \leftarrow Classifier(\mathbf{H}_f)$;

---

**Header–Payload Splitting and Feature Mapping.** From $\mathbf{S}^{(i)\prime}$, we retain $m \leq m_i$ packets and split each packet into header and payload. These bytes are mapped to feature matrices

$$\mathbf{X}_{\text{hdr}}^{(i)} \in \mathbb{R}^{m \times d_{\text{hdr}}}, \quad \mathbf{X}_{\text{pld}}^{(i)} \in \mathbb{R}^{m \times d_{\text{pld}}}, \quad \mathbf{X}^{(i)} = \text{Concat}(\mathbf{X}_{\text{hdr}}^{(i)}, \mathbf{X}_{\text{pld}}^{(i)}). \tag{17}$$

**Flow Embedding Structure.** Denote $\mathbf{X}^{(i)} = [\mathbf{x}_1^{(i)}, \ldots, \mathbf{x}_m^{(i)}]^\top$, where vertical slices trace inter-packet dynamics and horizontal slices encode intra-packet semantics.

**Flow-Level Encoding.** The adapter $\phi_{\text{adapter}}$ maps $\mathbf{X}^{(i)}$ into a fixed embedding:

$$\mathbf{H}_{\text{flow}}^{(i)} = \phi_{\text{adapter}}(\mathbf{X}^{(i)}) \in \mathbb{R}^{d_{\text{flow}}}. \tag{18}$$

We implement $\phi_{\text{adapter}}$ with several options: 1D-CNN, 2D-CNN, Linear projection, or ResNet blocks for hierarchical feature extraction.

## D.1 ADAPTER IMPLEMENTATIONS

For completeness, we provide the explicit mathematical forms of the four adapter modules described in Sec. 3.1.

- **1D-CNN:** with window size $r$ and weight $W \in \mathbb{R}^{r \times (d_h + d_p) \times d'}$, the feature map is

$$\mathbf{h}_j^{(i)} = \sigma\left(W * \mathbf{X}_{j:j+r,:}^{(i)}\right), \ \mathbf{h}_j^{(i)} \in \mathbb{R}^{d'}.$$

- **2D-CNN:** regarding $\mathbf{X}^{(i)} \in \mathbb{R}^{m \times (d_h + d_p)}$ as a 2D grid, with kernel $K \in \mathbb{R}^{r_m \times r_d}$ the feature map is

$$\mathbf{H}_{u,v}^{(i)} = \sigma \left( \sum_{a=1}^{r_m} \sum_{b=1}^{r_d} K_{a,b} \, \mathbf{X}_{u+a,v+b}^{(i)} \right), \ \mathbf{H}^{(i)} \in \mathbb{R}^{m' \times d'}.$$

- **Linear:** projection with $W \in \mathbb{R}^{d' \times m(d_h + d_p)}$ yields

$$\mathbf{H}_{\text{flow}}^{(i)} = W \cdot \text{vec}(\mathbf{X}^{(i)}), \ \mathbf{H}_{\text{flow}}^{(i)} \in \mathbb{R}^{d'}.$$

- **ResNet:** stacking convolutional blocks $F_\ell(\cdot)$ with residual connections gives

$$\mathbf{h}_{\ell+1}^{(i)} = F_\ell(\mathbf{h}_\ell^{(i)}) + \mathbf{h}_\ell^{(i)}, \ \mathbf{h}_\ell^{(i)} \in \mathbb{R}^{d'}.$$

### D.2 ITERATION OF IHL

At each iteration $t$, the embeddings of flow nodes $[\mathbf{H}_{\text{flow}}]_t$ and hyperedges $[\mathbf{H}_{\text{he}}]_t$ are updated dynamically via the following attention-based updates:

$$[\mathbf{H}_{\text{flow}}]_{t+1} = \sigma \left( \sum_j \frac{\exp\left([\mathbf{Q}_i]_t \mathbf{K}_j^T\right)}{\sum_k \exp\left([\mathbf{Q}_i]_t \mathbf{K}_k^T\right)} \mathbf{V}_j \right), \quad [\mathbf{H}_{\text{he}}]_{t+1} = \sigma \left( \sum_i \frac{\exp\left([\mathbf{Q}_j]_t \mathbf{K}_i^T\right)}{\sum_k \exp\left([\mathbf{Q}_j]_t \mathbf{K}_k^T\right)} \mathbf{V}_i \right).$$

(19)

In our framework, the implicit hyperedge embeddings are instantiated as a set of learnable, randomly initialized vectors whose incident nodes correspond to the semantic nodes extracted by the adapter, and whose connection weights are given by the computed hypergraph attention scores. By iteratively applying the above updates, the entire hypergraph evolves toward a stable configuration that minimizes the training loss, and prior work (Choudhuri et al. (2025)) has shown that the existence of such a convergent state is guaranteed.

Here, $\mathbf{Q}$ and $\mathbf{K}$ are the query and key vectors for the flow nodes and hyperedges, and $\sigma(\cdot)$ represents a non-linear activation function. The final converged flow semantic embeddings, denoted as $\mathbf{H}'_{\text{flow}}$, are derived.

## E DETAILED FORMULATION OF HIERARCHICAL CLASSIFICATION LOSS

In this appendix, we provide the full mathematical details of the hierarchical classification loss for large-scale ETC.

### E.1 COARSE- AND FINE-LEVEL LOGITS

Given fused flow embeddings $\mathbf{H}_{\text{HBG}} \in \mathbb{R}^{|\mathcal{V}_{\text{flow}}| \times d_F}$, the coarse- and fine-level logits are computed as:

$$\mathbf{z}_c = \mathbf{H}_{\text{HBG}} \mathbf{W}_c + \mathbf{b}_c, \quad \mathbf{z}_f = \mathbf{H}_{\text{HBG}} \mathbf{W}_f + \mathbf{b}_f,$$

(20)

where $\mathbf{W}_c \in \mathbb{R}^{d_F \times N_c}$, $\mathbf{W}_f \in \mathbb{R}^{d_F \times N_f}$ are learnable weights, and $\mathbf{b}_c, \mathbf{b}_f$ are biases.

### E.2 COARSE AND CONDITIONAL FINE PROBABILITIES

The coarse-level prediction probability is

$$p_c^{(n)}(i) = \text{softmax}(\mathbf{z}_c^{(n)})_i.$$

(21)

The conditional fine-level probability given coarse class $i$ is

$$p_{f|c}^{(n)}(j|i) = \frac{\exp(z_f^{(n),j}/\tau)}{\sum_{k:M_{ik}=1} \exp(z_f^{(n),k}/\tau)}, \quad \text{for } M_{ij} = 1,$$

(22)

where $\tau$ is a temperature hyperparameter, and $M \in \{0,1\}^{N_c \times N_f}$ encodes the coarse-fine mapping.

The marginalized fine-level probability is then

$$p_f^{(n)}(j) = \sum_{i=1}^{N_c} p_c^{(n)}(i) \, p_{f|c}^{(n)}(j|i).$$

(23)

Table 3: Performance comparison on three datasets (ACC range and improvement vs. baseline, %). Upward arrow (↑) indicates improvement over baseline.

| Type | CIC-IoT2023 | | ISCX-VPN2016 | | USTC-TFC2016 | |
|---|---|---|---|---|---|---|
| | ACC (%) | ↑ (%) | ACC (%) | ↑ (%) | ACC (%) | ↑ (%) |
| Sequence-based | 62.4–70.0 | 20.1 | 72.6–80.1 | 17.8 | 86.7–97.1 | 2.4 |
| Image-based | 72.3 | 14.3 | 91.2 | 4.2 | 97.6 | 0.5 |
| TFE-GNN, MH-GNN | 44.5–48.1 | 38.4–42.0 | 86.2–88.0 | 7.6–9.7 | 87.1–96.5 | 1.6–11.4 |
| E-GraphSAGE | 77.7 | 8.8 | 76.4 | 19.7 | 91.1 | 7.1 |
| $S^2$-ETR (best variant vs avg) | **86.5** | **8.8** | **95.4** | **4.2** | **98.1** | **0.5** |

### E.3 HIERARCHICAL CLASSIFICATION LOSS WITH BILINEAR REGULARIZATION

Finally, the hierarchical loss combines coarse- and fine-level supervision and optional bilinear regularization:

$$\mathcal{L}_{\text{hier}} = -\frac{1}{|\mathcal{V}_{\text{flow}}|} \sum_{n=1}^{|\mathcal{V}_{\text{flow}}|} \left[ \sum_{i=1}^{N_c} y_c^{(n),i} \log p_c^{(n)}(i) + \lambda \sum_{j=1}^{N_f} y_f^{(n),j} \log p_f^{(n)}(j) \right]$$
$$+ \gamma \frac{1}{|\mathcal{V}_{\text{flow}}|} \sum_{n=1}^{|\mathcal{V}_{\text{flow}}|} \sum_{i=1}^{N_c} \sum_{j=1}^{N_f} M_{ij} \left( (\mathbf{h}_n^{\text{fused}})^\top \mathbf{W}_h \mathbf{v}_{ij} - \log p_f^{(n)}(j) \right)^2, \tag{24}$$

where $\mathbf{h}_n^{\text{fused}}$ is the $n$-th fused flow embedding, $\mathbf{v}_{ij} \in \mathbb{R}^{d_F}$ is a learnable embedding for coarse-fine pair $(i, j)$, and $\lambda, \gamma$ are balancing hyperparameters.

This detailed formulation ensures proper hierarchical supervision while enforcing consistency between coarse and fine predictions via the bilinear term.

## F DETAILS OF EXPERIMENTS

## F.1 SCALE STUDY

Table 4: **Adapter performance (ACC / PRE / REC / F1) across data ratios.** Left block: 10%–50%. Right block: 60%–100%. IP_num and Flow_num are shown per row (same for all adapters under the same scale).

| Scale | Adapter | IP_num | Flow_num | ACC | PRE | REC | F1 | Scale | Adapter | IP_num | Flow_num | ACC | PRE | REC | F1 |
|---|---|---|---|---|---|---|---|---|---|---|---|---|---|---|---|
| | | | 10% – 50% | | | | | | | | 60% – 100% | | | | |
| 10% | $S^2$-ETR (1D-CNN) | 1435 | 8642 | 0.7831 | 0.7015 | 0.6838 | 0.6873 | 60% | $S^2$-ETR (1D-CNN) | 3269 | 51853 | 0.8693 | 0.8084 | 0.8318 | 0.8195 |
| | $S^2$-ETR (ResNet) | 1435 | 8642 | 0.7848 | 0.7013 | 0.7082 | 0.7025 | | $S^2$-ETR (ResNet) | 3269 | 51853 | 0.8767 | 0.8132 | 0.8467 | 0.8285 |
| | $S^2$-ETR (2D-CNN) | 1435 | 8642 | 0.7895 | 0.7058 | 0.7678 | 0.7295 | | $S^2$-ETR (2D-CNN) | 3269 | 51853 | 0.8205 | 0.7469 | 0.8300 | 0.7749 |
| | $S^2$-ETR (Linear) | 1435 | 8642 | 0.7721 | 0.6874 | 0.6809 | 0.6822 | | $S^2$-ETR (Linear) | 3269 | 51853 | 0.8796 | 0.8125 | 0.8641 | 0.8354 |
| 20% | $S^2$-ETR (1D-CNN) | 2017 | 17284 | 0.8282 | 0.7670 | 0.7807 | 0.7724 | 70% | $S^2$-ETR (1D-CNN) | 3466 | 60496 | 0.8685 | 0.8085 | 0.8411 | 0.8218 |
| | $S^2$-ETR (ResNet) | 2017 | 17284 | 0.8403 | 0.7834 | 0.8006 | 0.7898 | | $S^2$-ETR (ResNet) | 3466 | 60496 | 0.8796 | 0.8164 | 0.8563 | 0.8338 |
| | $S^2$-ETR (2D-CNN) | 2017 | 17284 | 0.8134 | 0.7280 | 0.7961 | 0.7559 | | $S^2$-ETR (2D-CNN) | 3466 | 60496 | 0.8331 | 0.7601 | 0.8422 | 0.7897 |
| | $S^2$-ETR (Linear) | 2017 | 17284 | 0.8478 | 0.8040 | 0.7952 | 0.7989 | | $S^2$-ETR (Linear) | 3466 | 60496 | 0.8778 | 0.8087 | 0.8601 | 0.8313 |
| 30% | $S^2$-ETR (1D-CNN) | 2417 | 25926 | 0.8423 | 0.7856 | 0.7888 | 0.7854 | 80% | $S^2$-ETR (1D-CNN) | 3634 | 69138 | 0.8793 | 0.8231 | 0.8446 | 0.8330 |
| | $S^2$-ETR (ResNet) | 2417 | 25926 | 0.8459 | 0.7811 | 0.7912 | 0.7851 | | $S^2$-ETR (ResNet) | 3634 | 69138 | 0.8883 | 0.8249 | 0.8751 | 0.8472 |
| | $S^2$-ETR (2D-CNN) | 2417 | 25926 | 0.8000 | 0.7205 | 0.7913 | 0.7457 | | $S^2$-ETR (2D-CNN) | 3634 | 69138 | 0.8434 | 0.7714 | 0.8515 | 0.8021 |
| | $S^2$-ETR (Linear) | 2417 | 25926 | 0.8575 | 0.7983 | 0.8107 | 0.8032 | | $S^2$-ETR (Linear) | 3634 | 69138 | 0.8857 | 0.8214 | 0.8726 | 0.8442 |
| 40% | $S^2$-ETR (1D-CNN) | 2772 | 34569 | 0.8583 | 0.8012 | 0.8118 | 0.8046 | 90% | $S^2$-ETR (1D-CNN) | 3774 | 77780 | 0.8792 | 0.8342 | 0.8309 | 0.8317 |
| | $S^2$-ETR (ResNet) | 2772 | 34569 | 0.8593 | 0.7875 | 0.8283 | 0.8060 | | $S^2$-ETR (ResNet) | 3774 | 77780 | 0.8959 | 0.8368 | 0.8772 | 0.8550 |
| | $S^2$-ETR (2D-CNN) | 2772 | 34569 | 0.8101 | 0.7314 | 0.8110 | 0.7457 | | $S^2$-ETR (2D-CNN) | 3774 | 77780 | 0.8844 | 0.8324 | 0.8474 | 0.8391 |
| | $S^2$-ETR (Linear) | 2772 | 34569 | 0.8639 | 0.7977 | 0.8290 | 0.8116 | | $S^2$-ETR (Linear) | 3774 | 77780 | 0.8954 | 0.8317 | 0.8889 | 0.8572 |
| 50% | $S^2$-ETR (1D-CNN) | 3041 | 43211 | 0.8632 | 0.8050 | 0.8085 | 0.8062 | 100% | $S^2$-ETR (1D-CNN) | 3891 | 86423 | 0.8867 | 0.8328 | 0.8530 | 0.8422 |
| | $S^2$-ETR (ResNet) | 3041 | 43211 | 0.8720 | 0.8039 | 0.8471 | 0.8228 | | $S^2$-ETR (ResNet) | 3891 | 86423 | 0.9009 | 0.8548 | 0.8815 | 0.8665 |
| | $S^2$-ETR (2D-CNN) | 3041 | 43211 | 0.8153 | 0.7532 | 0.8243 | 0.7755 | | $S^2$-ETR (2D-CNN) | 3891 | 86423 | 0.9016 | 0.8463 | 0.9004 | 0.8705 |
| | $S^2$-ETR (Linear) | 3041 | 43211 | 0.8716 | 0.8050 | 0.8453 | 0.8223 | | $S^2$-ETR (Linear) | 3891 | 86423 | 0.9089 | 0.8554 | 0.8981 | 0.8749 |

## F.2 ROBUSTNESS STUDY

Table 5: **Adapter comparison on three datasets.** Top block: ACC (best-run, mean, variance); bottom block: F1 (best-run, mean, variance).

| Adapter | ISCX-VPN2016 | | | USTC-TFC2016 | | | CIC-IoT2023 | | |
|---|---|---|---|---|---|---|---|---|---|
| | ACC | Mean | Var | ACC | Mean | Var | ACC | Mean | Var |
| | | | | *ACC (best-run / mean$\pm\sqrt{var}$)* | | | | | |
| $S^2$-ETR (2D-CNN) | 0.9438 | 0.9344 | 3.52e-05 | 0.9506 | 0.9410 | 2.29e-05 | 0.8650 | 0.8374 | 1.74e-04 |
| $S^2$-ETR (ResNet) | 0.9537 | 0.9391 | 5.46e-05 | 0.9812 | 0.9736 | 1.21e-05 | 0.8612 | 0.8357 | 2.90e-04 |
| $S^2$-ETR (Linear) | 0.9460 | 0.9322 | 2.45e-05 | 0.9706 | 0.9604 | 1.93e-05 | 0.8413 | 0.8222 | 1.10e-04 |
| $S^2$-ETR (1D-CNN) | 0.9405 | 0.9248 | 5.74e-05 | 0.9794 | 0.9691 | 2.23e-05 | 0.8413 | 0.8140 | 1.87e-04 |
| | | | | *F1 (best-run / mean$\pm\sqrt{var}$)* | | | | | |
| $S^2$-ETR (2D-CNN) | 0.9406 | 0.9297 | 6.17e-05 | 0.9501 | 0.9404 | 2.61e-05 | 0.8652 | 0.8376 | 1.73e-04 |
| $S^2$-ETR (ResNet) | 0.9533 | 0.9325 | 1.49e-04 | 0.9812 | 0.9734 | 1.21e-05 | 0.8609 | 0.8357 | 1.70e-04 |
| $S^2$-ETR (Linear) | 0.9342 | 0.9164 | 1.20e-04 | 0.9706 | 0.9603 | 2.01e-05 | 0.8418 | 0.8221 | 1.14e-04 |
| $S^2$-ETR (1D-CNN) | 0.9260 | 0.9094 | 1.47e-04 | 0.9793 | 0.9690 | 2.26e-05 | 0.8404 | 0.8133 | 1.85e-04 |

Table 6: **Efficiency comparison on USTC-TFC2016.** For each model we report inference time (ms, mean$\pm\sqrt{\text{var}}$) and peak CPU memory (MB).

| Model | Inference Time (ms) | Peak Memory (MB) |
|---|---|---|
| *Baselines* | | |
| 2D-CNN | 0.5340±0.0070 | 17.57 |
| FlowPic | 0.1056±0.0214 | 74.99 |
| Hast | 0.0790±0.0069 | 13.11 |
| TSCRNN | 0.7267±0.0084 | 15.01 |
| *$S^2$-ETR variants* | | |
| $S^2$-ETR (2D-CNN) | 0.0432±0.0069 | 42.00 |
| $S^2$-ETR (ResNet) | 0.0982±0.0026 | 49.13 |
| $S^2$-ETR (Linear) | 0.0179±0.0051 | 41.18 |
| $S^2$-ETR (1D-CNN) | 0.0282±0.0035 | 34.68 |

## F.3 COMPLEXITY

Table 7: **Complexity comparison of different models.** We report the number of parameters and FLOPs for each model.

| Model | Parameters | FLOPs |
|---|---|---|
| *Main models* | | |
| $S^2$-ETR (2D-CNN) | 437.834K | 12.112M |
| $S^2$-ETR (ResNet) | 3.320M | 89.962M |
| $S^2$-ETR (1D-CNN) | 1.031M | 8.850M |
| $S^2$-ETR (Linear) | 3.025M | 3.935M |
| *Baselines* | | |
| 1D-CNN | 344.234K | 7.251M |
| 2D-CNN | 191.274K | 4.834M |
| FlowPic | 320.720K | 636.700K |
| HAST | 24.162K | 431.360K |
| TSCRNN | 215.050K | 18.516M |
| Netmamba | 1.859M | 2.806M |

## F.4 LARGE-SCALE AND HIERARCHICAL CLASSIFICATION

Table 8: **Performance comparison on the large dataset (43 Class).**

| Model | Accuracy | Precision | Recall | F1-Score |
|---|---|---|---|---|
| 1D-CNN | 0.1474 | 0.1561 | 0.1474 | 0.1453 |
| 2D-CNN | 0.1284 | 0.1274 | 0.1284 | 0.1230 |
| FlowPic | 0.1567 | 0.1558 | 0.1567 | 0.1528 |
| HAST | 0.1381 | 0.1403 | 0.1381 | 0.1326 |
| TSCRNN | 0.1040 | 0.1021 | 0.1040 | 0.0994 |
| NetMamba | 0.4134 | 0.4080 | 0.4134 | 0.4018 |
| *$S^2$-ETR variants* | | | | |
| $S^2$-ETR (none+Linear) | 0.0930 | 0.0539 | 0.0930 | 0.0522 |
| $S^2$-ETR (hier+Linear) | 0.3749 | 0.3784 | 0.3642 | 0.3656 |
| $S^2$-ETR (hier+1D-CNN) | 0.4174 | 0.4117 | 0.4039 | 0.4038 |
| $S^2$-ETR (hier+ResNet) | 0.3819 | 0.3814 | 0.3763 | 0.3738 |

Table 9: **Hierarchical classification results on both coarse (5 Class) and fine (43 Class) levels.**

| Model | Fine (43 Class) | | | | Coarse (5 Class) | |
|---|---|---|---|---|---|---|
| | Acc | Pre | Rec | F1 | Acc | F1 |
| $S^2$-ETR (hier+Linear) | 0.3749 | 0.3784 | 0.3642 | 0.3656 | 0.6698 | 0.5719 |
| $S^2$-ETR (hier+1D-CNN) | 0.4174 | 0.4117 | 0.4039 | 0.4038 | 0.6955 | 0.6011 |
| $S^2$-ETR (hier+ResNet) | 0.3819 | 0.3814 | 0.3763 | 0.3738 | 0.6683 | 0.5735 |

## G ABLATION STUDY

## G.1 ABLATION OF COMPONENTS

Table 10: **Ablation results under the same style.** Only best-run metrics are reported (Accuracy / Precision / Recall / F1-Score).

| Mode | CIC-IoT2023 | | | | ISCX-VPN2016 | | | | USTC-TFC2016 | | | |
|---|---|---|---|---|---|---|---|---|---|---|---|---|
| | ACC | Pre | Rec | F1 | ACC | Pre | Rec | F1 | ACC | Pre | Rec | F1 |
| Full model | 0.8650 | 0.8667 | 0.8650 | 0.8652 | 0.9438 | 0.9264 | 0.9454 | 0.9329 | 0.9506 | 0.9545 | 0.9506 | 0.9501 |
| W/o topology branch | 0.4912 | 0.5053 | 0.4871 | 0.4769 | 0.5771 | 0.5253 | 0.5053 | 0.4969 | 0.7456 | 0.7650 | 0.7327 | 0.7001 |
| W/o semantic branch | 0.8438 | 0.8463 | 0.8405 | 0.8403 | 0.9262 | 0.9352 | 0.9299 | 0.9306 | 0.9538 | 0.9569 | 0.9519 | 0.9521 |
| W/o IHL | 0.4000 | 0.4711 | 0.3990 | 0.3876 | 0.4350 | 0.3380 | 0.3813 | 0.3525 | 0.6469 | 0.6064 | 0.6420 | 0.6023 |

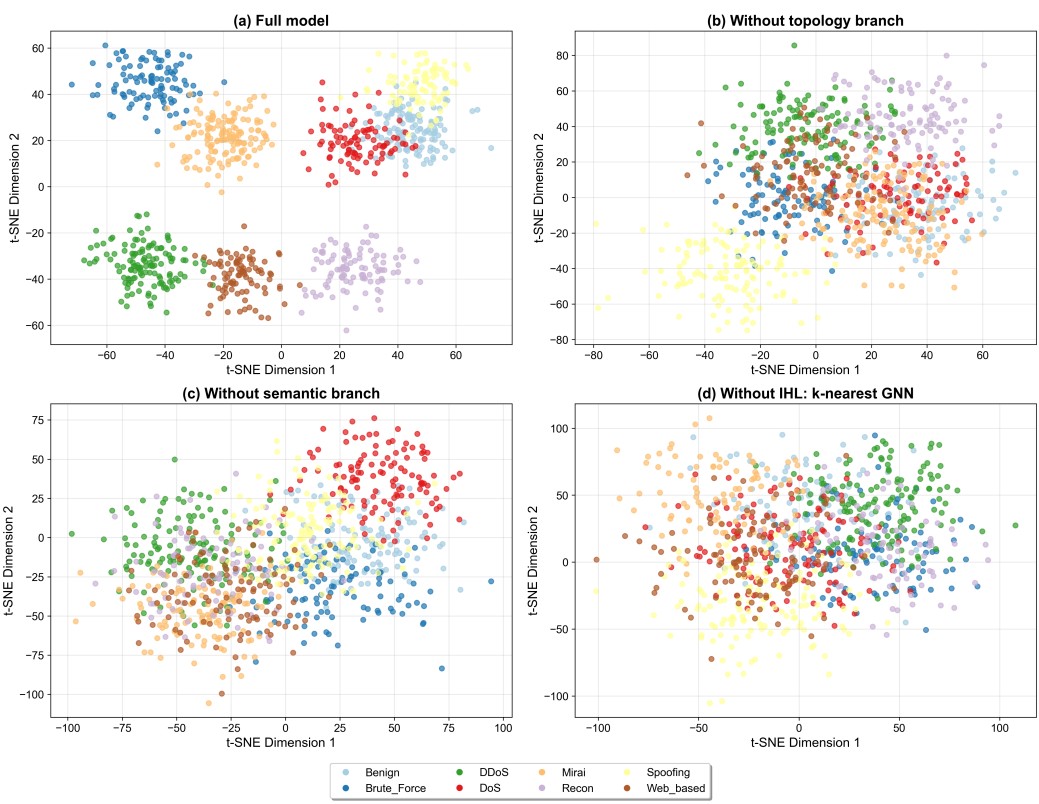

Figure 6: T-SNE ablation result. Specifically, k-nearest GNN is used as a comparison for the implicit hypergraph learning, which establishes fixed connections between k semantically similar nodes at the start of the experiment using the k-nearest neighbor algorithm.

## G.2 COMPARISON OF METHODS FOR MITIGATING OVER-SQUASHING

Table 11: Performance comparison of Mitigating over-squashing methods on CipherSpectrum2025.

| Variants | ACC | PRE | REC | F1 |
|---|---|---|---|---|
| SRDF | 0.9751 | 0.9761 | 0.9751 | 0.9750 |
| ComFy | 0.9470 | 0.9533 | 0.9470 | 0.9451 |
| ours (IHL) | **0.9780** | **0.9805** | **0.9780** | **0.9779** |

## G.3 STATIC AND DYMANIC FUSION COMPARISON

Table 12: Performance comparison of static and dynamic fusion on CipherSpectrum2025.

| Variants | ACC | PRE | REC | F1 | best_epoch | inference_time (ms) |
|---|---|---|---|---|---|---|
| GMU | 0.9419 | 0.9489 | 0.9419 | 0.9400 | 23 | $1.71 \pm 0.03$ |
| AFF | 0.9432 | 0.9511 | 0.9432 | 0.9407 | 27 | $1.83 \pm 0.27$ |
| ours (static) | **0.9780** | **0.9805** | **0.9780** | **0.9779** | **102** | $\mathbf{1.00 \pm 0.05}$ |

