# OpenReview forum: "S$^2$-ETC: Semantic-Structure Aware Encrypted Traffic Classification via Hyper-Bipartite Graph"
_ICLR.cc/2026/Conference — Submitted to ICLR 2026_

### Official Review · Reviewer_T3wc · 2025-10-16

**Soundness:** 2
**Presentation:** 2
**Contribution:** 2
**Rating:** 4
**Confidence:** 3

**Summary:**

This paper proposes S²-ETR, a novel framework for encrypted traffic classification that integrates semantic and topological features via a Hyper-Bipartite Graph (HBG). The method consists of an adapter, an HBG module, and a hierarchical classifier, and is evaluated on four datasets, showing superior performance over 11 baselines.

**Strengths:**

- Comprehensive experiments across multiple datasets, scales, and scenarios;

- Strong performance in efficiency and scalability, especially in large-scale hierarchical classification.

**Weaknesses:**

- [Mandatory] Many statements in the Introduction require additional references or other forms of support, such as the over-squashing problem in topology-based GNNs.

- [Mandatory] Anonymization (e.g., IP/port replacement) is a strong prior and may introduce bias.

- [Mandatory] Some definitions referenced in Section 3 are supposed to be provided in Section 2, but this part of the definitions is missing in Section 2.

- [Mandatory] The theoretical analysis is based on MPNN, but its connection to the attention-based IHL used in the model is not clearly established;

- [Mandatory] Key implementation aspects of IHL (e.g., initialization, convergence criteria) are omitted;

- [Mandatory] The fusion weight $\alpha$ is heuristic and lacks theoretical or empirical justification;

**Questions:**

Please refer to Weaknesses. Btw, I have some optional questions:

- [Optional] How are hyperedges initialized in IHL, and did you encounter non-convergence during iterative updates?

- [Optional] Were the GNN baselines ensured to have graph construction comparable to the topological branch of HBG?

- [Optional] Could anonymization discard discriminative semantic patterns (e.g., port-based features), thus limiting performance?

---

> ### Author Response · Authors · 2025-11-22
>
> ### Weakness 1:
> Many statements in the Introduction require additional references or other forms of support, such as the over-squashing problem in topology-based GNNs.
>
> ### Response to Weakness 1:
>
> We thank the reviewer for pointing this out. We agree that the current Introduction does not provide sufficient discussion and supporting evidence for several key background issues, such as the over-squashing problem of GNNs on high-order topological structures and the motivation for adopting a topology-based approach to encrypted traffic classification.
>
> In the revised manuscript, we expand the Introduction and Related Work (Sections 1 and 2) to (i) summarize the limitations of traditional machine learning methods that rely on handcrafted statistical features, which are hard to tune and generalize poorly to new encryption schemes and application behaviors; (ii) highlight that sequence-based and image-based deep models mainly capture per-flow byte-sequence semantics, while recent studies [1][2] show that carefully crafted malicious flows can be made almost indistinguishable from benign ones after encryption, with highly randomized, high-entropy byte sequences that are hard to separate at the single-flow level; and (iii) motivate a shift from modeling isolated flows to exploiting relations among flows via topology-based analyses. We further clarify that, in traffic interaction graphs, most discriminative information lies on edges rather than nodes, and that message passing through intermediate nodes can exacerbate over-squashing and scalability issues, especially when multiple flows between the same IP pair are collapsed into a single edge embedding. Based on this analysis, we explicitly state that our goal is to jointly model intra-flow semantics and inter-flow relations so that topology complements single-flow models and yields more accurate and more generalizable detection across diverse encrypted-traffic scenarios. To better convey this motivation and the over-squashing issue, we add Fig. 1 and the corresponding discussion in the Introduction and restructure Section 2 (Related Work), with all newly added content highlighted in blue.
>
> [1] *Zhuoqun Fu, Mingxuan Liu, Yue Qin, Jia Zhang, Yuan Zou, Qilei Yin, Qi Li, and Haixin Duan. Encrypted malware traffic detection via graph-based network analysis. In Proceedings of the 25th International Symposium on Research in Attacks, Intrusions and Defenses, pp. 495–509, 2022.
> [2] *Chuanpu Fu, Qi Li, and Ke Xu. Detecting unknown encrypted malicious traffic in real time via flow interaction graph analysis. In 30th Annual Network and Distributed System Security Symposium, NDSS 2023
>
> ### Weakness 2:
> Anonymization (e.g., IP/port replacement) is a strong prior and may introduce bias.
>
> ### Response to Weakness 2:
> We appreciate this important comment. Our use of IP/port anonymization is mainly motivated by two considerations:
>
> (1)**Mitigating environment-specific overfitting.**
> Prior work [3] has shown that models may overfit to dataset-specific artifacts (e.g., particular port numbers or IP ranges) instead of learning robust behavioral patterns. By removing such explicit identifiers, we aim to reduce environment-specific bias and improve generalization across different network environments.
>
> (2)**Consistency with common practice.**
> Anonymization is also a widely adopted setting in encrypted traffic classification. Several strong baselines, such as YaTC, perform similar processing by masking IP addresses and ports (e.g., replacing them with zeros). We follow this practice to ensure fair and comparable evaluation.
>
> [3]*Nimesha Wickramasinghe, Arash Shaghaghi, Gene Tsudik, and Sanjay Jha. Sok: Decoding the
> enigma of encrypted network traffic classifiers. In Proceedings of the IEEE Symposium on Secu-
> rity and Privacy, pp. 1825–1843. IEEE, 2025.
>
>
> ### Weakness 3:
>
> Some definitions referenced in Section 3 are supposed to be provided in Section 2, but this part of the definitions is missing in Section 2.
>
> ### Response to Weakness 3:
>
> We thank the reviewer for pointing out this structural issue. Indeed, Section 3 originally referred to a number of concepts and notations that were not clearly defined beforehand. In the revision, we reorganize Sections 2 and 3 and move the relevant definitions and formulas to a dedicated subsection and the appendix, with proper cross-references from Section 3. This ensures that all symbols and concepts are introduced before they are used in the model and theoretical analysis. The revised parts are highlighted in blue.

---

> > ### Author Response · Authors · 2025-11-22
> >
> > ### Weakness 4:
> >
> > The theoretical analysis is based on MPNN, but its connection to the attention-based IHL used in the model is not clearly established.
> >
> > ### Response to Weakness 4:
> >
> > Our implicit hypergraph learning module (IHL) can in fact be viewed as a specific instance of a Message Passing Neural Network (MPNN), where the hypergraph attention scores act as dynamically learned edge (hyperedge) weights. Existing works [4][5] on implicit hypergraph learning have shown that such attention-weighted implicit hypergraphs can effectively capture long-range dependencies and thereby mitigate over-squashing induced by limited edge connectivity. We emphasize this connection more clearly in the revised manuscript and explicitly state how the attention-based IHL fits into the MPNN framework. The corresponding clarifications are highlighted in blue.
> >
> > [4] *Gu F, Chang H, Zhu W, et al. Implicit graph neural networks[J]. Advances in neural information processing systems, 2020, 33: 11984-11995.
> > [5] *Choudhuri A, Zhong Y, Adhikari B. Implicit hypergraph neural network[J]. arXiv preprint arXiv:2508.14101, 2025.
> >
> > ### Weakness 5:
> > Key implementation aspects of IHL (e.g., initialization, convergence criteria) are omitted.
> >
> > ### Response to Weakness 5:
> >
> > We thank the reviewer for pointing out this omission. In the revised manuscript, we add a dedicated subsection in Appendix D.2 that details the implementation of implicit hypergraph learning (IHL).
> >
> > Concretely, the implicit hyperedges are parameterized as learnable vectors that are randomly initialized, with dimensionality matching the semantic flow representations produced by the adapter. Each hyperedge is connected to the semantic nodes via an attention mechanism, and the resulting hypergraph attention scores are used as edge weights. During training, node and hyperedge embeddings are jointly optimized, and we regard the model as converged when the training/validation loss reaches a stable minimum under our early-stopping criterion.
> >
> > Prior studies on implicit hypergraph and attention-based message passing have shown that such iterative updates converge to a stable state under standard optimization settings. Furthermore, in our experiments we did not observe any non-convergence issues.
> >
> > ### Weakness 6:
> >
> > The fusion weight is heuristic and lacks theoretical or empirical justification.
> >
> > ### Response to Weakness 6:
> >
> > We thank the reviewer for raising this important point. In the original submission, we adopt a statically computed coefficient α that quantifies the sparsity of the IP–Flow graph, and use it to balance feature fusion between the structural and semantic branches.
> >
> > To further justify this design, we introduce two dynamic fusion baselines—a gating mechanism and the AFF module—which are briefly described in Section 3 and evaluated in our experiments. These baselines allow us to directly compare static versus dynamic fusion in terms of both performance and inference cost. As reported in Table 12, our static fusion scheme achieves 97% accuracy, outperforming both the gating and AFF baselines by about 3.5%, while also reducing inference time by approximately 70%.
> >
> > Empirically, we observe that the learnable fusion weights in the dynamic variants tend to converge prematurely and get stuck in suboptimal local minima. In contrast, the static weight, which is designed to reflect the global topological sparsity of the interaction graph, provides a more stable trade-off between the two branches and leads to more consistent and superior performance in our setting.
> >
> > ### Question 1:
> > How are hyperedges initialized in IHL, and did you encounter non-convergence during iterative updates?
> >
> > ### Response to Question 1:
> > In IHL, each hyperedge is initialized as a learnable d-dimensional vector, where d is the dimensionality of the single-flow semantic representation extracted by the adapter. Through the attention mechanism, each hyperedge is, by design, connected to all semantic node vectors in a fully connected manner, and the resulting attention scores serve as edge weights. Prior work [4][5] on implicit hypergraphs has shown that such structures converge to a stable state, and in our experiments we did not observe non-convergence or instability during iterative updates.

---

> > > ### Author Response · Authors · 2025-11-22
> > >
> > > ### Question 2:
> > >
> > > Were the GNN baselines ensured to have graph construction comparable to the topological branch of HBG?
> > >
> > > ### Response to Question 2:
> > >
> > > Yes. For the GNN baselines, the underlying topology is a multi-edge graph. E-GraphSAGE compresses multiple edges between two nodes into a single edge, while ST-Graph (not publicly released) selects and retains only the most important edges based on an importance ranking. In contrast, our method promotes edges to independent nodes, forming a bipartite graph where IPs and flows are decoupled. This design preserves as much edge information as possible.
> > >
> > > From a communication perspective, our topological branch and the GNN baselines are comparable, since they operate on equivalent interaction structures. However, from an information-theoretic perspective, our construction retains richer edge-level information, which is then exploited by HBG.
> > >
> > > ### Question 3:
> > >
> > > Could anonymization discard discriminative semantic patterns (e.g., port-based features), thus limiting performance?
> > >
> > > ### Response to Question 3:
> > >
> > > Yes, anonymization can indeed remove some discriminative patterns. In our reproductions of several deep-learning baselines, we observed that anonymization leads to a 1–5% performance drop compared with the numbers reported in the original papers.
> > >
> > > However, we consider this sacrifice necessary: prior studies [6] have shown that modern attackers deliberately mix malicious and benign traffic patterns, making simple port- or IP-based rules unreliable. Anonymization forces models to rely on behavioral and statistical patterns rather than "shortcuts" [7], which is crucial for realistic deployment in real-world networks.
> > >
> > > [6] *Beitis A, Vanhoef M. Haunted by Legacy: Discovering and Exploiting Vulnerable Tunnelling Hosts[C]//34th USENIX Security Symposium (USENIX Security 25). 2025: 7135-7152.
> > >
> > > [7] *Nimesha Wickramasinghe, Arash Shaghaghi, Gene Tsudik, and Sanjay Jha. Sok: Decoding the
> > > enigma of encrypted network traffic classifiers. In Proceedings of the IEEE Symposium on Secu-
> > > rity and Privacy, pp. 1825–1843. IEEE, 2025.

---

### Official Review · Reviewer_FCWc · 2025-10-27

**Soundness:** 2
**Presentation:** 1
**Contribution:** 2
**Rating:** 2
**Confidence:** 3

**Summary:**

This paper presents a framework for encrypted traffic classification designed to address issues of overfitting and generalization. The proposed method is centered on a model, the S^2-ETR, which implements a Hyper-Bipartite Graph (HBG) structure.

The framework utilizes a dual-branch architecture to process both semantic and topological information from network traffic. For the semantic branch, the system preprocesses PCAP data by extracting the first six packets of each flow, consisting of an 80-byte header and a 240-byte payload. Within this branch, IP addresses and port numbers are anonymized. The resulting data is then passed to a configurable encoder, referred to as an adapter (e.g., 1D/2D CNN, ResNet, Linear), to generate a content-based representation. For the topological branch, the original, non-anonymized IP information is used to construct an IP-Flow bipartite graph. The model learns embeddings for the IP nodes within this graph to represent structural relationships.

The paper's theoretical analysis posits that the HBG structure is intended to mitigate the over-squashing problem common in graph neural networks. It is argued that this is achieved by replacing long, multi-hop information paths with constant-depth aggregation steps, thereby reducing long-range dependencies. The framework was evaluated on three datasets (CICIoT2023, ISCXVPN2016, and USTC-TFC2016) using the four different encoder configurations.

**Strengths:**

1. The main new idea in this work is the "Hyper-Bipartite Graph" (HBG), which is a framework with two parts. The first part (the topological branch) models the problem in a smart way by representing how IP addresses and data flows are connected. The second part (the semantic branch) uses an attention mechanism to cleverly learn more complex relationships.
2. The authors present and provide a mathematical proof explaining why their proposed HBG architecture is superior to traditional IP topology graphs.
3. The method was evaluated through comprehensive comparative experiments against eleven deep learning-based, pre-training-based, and graph neural network-based models across three distinct datasets. Some of the dataset choices are intriguing.

**Weaknesses:**

1. This paper is heavily packed with technical details and expresses many concepts and designs through mathematical formulas. However, most of these equations do not actually aid understanding; instead, they significantly reduce the readability of the paper. I suggest reducing the use of formulas in the main text. Rigorous definitions and proofs can be moved to the appendix, while the main body should use simpler and more intuitive representations such as figures.
2. The authors claim that S^2-ETR addresses the limitations of deep learning methods, pretraining-based methods, and topology-based GNNs. However, the paper only discusses improvements over topology-based GNNs. The authors should elaborate more on why a topology-based approach was adopted and what theoretical motivations underlie this choice, in order to strengthen the overall motivation.
3. A short overview at the beginning of Section 3—summarizing Figure 1’s workflow and core idea—would greatly improve readability and prepare readers for the detailed designs later on.
4. Accuracy and efficiency are claimed as major contributions, possibly explaining why the authors did not employ large models. However, as far as I know, the most efficient methods remain traditional machine-learning approaches such as FlowPrint and AppScanner. Even when using raw packet bytes as input, traditional models like XGBoost can still achieve high accuracy. It is therefore suggested that the authors include comparisons with such traditional baselines.
5. In the experiments, it would be useful to include comparisons with NetMamba, which likewise emphasizes efficiency. In addition, the reported accuracies on ISCX-VPN2016 are noticeably lower than those in the original papers (e.g., YATC and TrafficFormer). This may reflect differences in pretraining data or other factors; please report the exact pretraining datasets used and explain the lower accuracy. Finally, because the performance gap on USTC-TFC2016 is relatively small across methods, adding more datasets would help substantiate the generality of the conclusions.

**Questions:**

1. Add more discussion on the motivation for adopting a topology-based approach.
2. Restructure Sections 2 and 3 to present the architecture and key designs more intuitively, avoiding excessive reliance on formulas.
3. Include additional datasets and comparative experiments, such as with traditional machine-learning methods (FlowPrint, AppScanner, raw packet–based XGBoost), and NetMamba.

---

> ### Author Response · Authors · 2025-11-22
>
> ### **Weakness 1**:
> This paper is heavily packed with technical details and expresses many concepts and designs through mathematical formulas. However, most of these equations do not actually aid understanding; instead, they significantly reduce the readability of the paper. I suggest reducing the use of formulas in the main text. Rigorous definitions and proofs can be moved to the appendix, while the main body should use simpler and more intuitive representations such as figures.
>
> ### **Response to Weakness 1**:
> We thank the reviewer for pointing out that the excessive use of mathematical formulas in the paper affects readability. Following the suggestion, we have streamlined and restructured the main text, and all updated content is marked in blue:
>
> (1) **Restructuring Section 2 and Section 3**: For derivations that do not directly affect the understanding of the main storyline, such as the preliminaries and the IHL iterative formulas, we have moved them to the appendix, keeping in the main text only the key computational procedures and necessary notation. Section 2 is now Related Works, which presents the limitations of existing methods, emphasizes the motivation for combining topology and semantic information, and describes the oversquashing problem and corresponding solutions. Section 3 now starts with a comprehensive introduction to the framework diagram.
>
> (2) **Revising the introduction and methodology**: We have restructured the Introduction and Methodology sections, added Figure 1 (showing existing methods and their limitations) and Figure 2 (the modeling diagram), and marked the corresponding explanations in blue. These figures use a graphical form to illustrate the model structure, information flow, and core design ideas.
>
> We believe these adjustments can significantly improve the overall readability of the paper while maintaining the rigor of the method description. The more complete derivations and proofs have been collected in the appendix for interested readers to consult.
>
> ### Weakness 2:
> The authors claim that S^2-ETR addresses the limitations of deep learning methods, pretraining-based methods, and topology-based GNNs. However, the paper only discusses improvements over topology-based GNNs. The authors should elaborate more on why a topology-based approach was adopted and what theoretical motivations underlie this choice, in order to strengthen the overall motivation.
>
> ### Response to Weakness 2:
>
> We thank the reviewer for pointing out the need to better motivate our topology-based design. In the revised manuscript, we have substantially expanded the discussion in both the Introduction and Related Work sections. Specifically, we highlight recent findings showing that, under strong encryption, carefully crafted malicious flows can become almost indistinguishable from benign ones at the single-flow level: their byte sequences are highly randomized and exhibit high entropy, which severely limits the discriminative power of sequence- or image-based deep models and pretraining-based methods that operate solely on individual flows. Motivated by these observations, we argue that encrypted traffic classification (ETC) should shift from modeling isolated flows to exploiting relations among flows, which is naturally captured by topology-based analyses.
>
> We further clarify that, in traffic interaction graphs, most discriminative information resides on edges (i.e., interactions between flows) rather than nodes. We discuss how conventional topology-based GNNs suffer from over-squashing and scalability issues when information must traverse multiple intermediate nodes, especially when multiple flows between the same IP pair are collapsed into a single edge embedding. These limitations partly explain the limited adoption of existing topology-based approaches for highly encrypted, large-scale traffic. Building on these observations, we position $S^2$-ETR as a framework that jointly models intra-flow semantics and inter-flow relations, so that topology serves as a complement, rather than a replacement, to deep or pretraining-based semantic modeling. We believe this expanded discussion strengthens both the theoretical and practical motivation for adopting a topology-aware approach in the encrypted traffic setting.

---

> > ### Author Response · Authors · 2025-11-22
> >
> > ### Weakness 3:
> > A short overview at the beginning of Section 3—summarizing Figure 1’s workflow and core idea—would greatly improve readability and prepare readers for the detailed designs later on.
> >
> > ### Response to Weakness 3:
> > We fully agree with the reviewer’s suggestion to add an overall overview. According to this comment, we have added an overview subsection (Section 3.1 FRAMEWORK OF THE PROPOSED S2-ETR) at the beginning of Section 3, whose main contents are as follows: we use three paragraphs to explain the framework diagram in Figure 3, outlining the complete pipeline from raw traffic to feature extraction, topology construction, semantic aggregation, and final classification output; we list the three core design components of S²-ETR, namely the topology branch, the semantic branch, and the fusion module, and use 1–2 sentences to explain the specific problem that each component is designed to address. We believe these revisions can significantly alleviate the abrupt feeling of “directly entering technical details” and make it easier for readers to grasp the main storyline when reading Section 3.
> >
> > ### Weakness 4:
> > Accuracy and efficiency are claimed as major contributions, possibly explaining why the authors did not employ large models. However, as far as I know, the most efficient methods remain traditional machine-learning approaches such as FlowPrint and AppScanner. Even when using raw packet bytes as input, traditional models like XGBoost can still achieve high accuracy. It is therefore suggested that the authors include comparisons with such traditional baselines.
> >
> > ### Response to Weakness 4:
> > We thank the reviewer for the suggestion regarding comparisons with traditional methods, which is very important for evaluating the contribution. In the revised manuscript, we have made the following additions:
> >
> > We have added ML-based classification by including traditional machine-learning baselines. Specifically, we have supplemented comparative experiments with FlowPrint, AppScanner, and XGBoost using raw or simple statistical features as input, and the results are shown in Table 2. On the VPN weakly encrypted application classification task, these three ML-based methods can achieve 72%–90% accuracy; however, on highly encrypted datasets such as CICIoT2023 and CipherSpectrum2025, their performance degrades significantly. In particular, FlowPrint reaches only 10% accuracy on CipherSpectrum2025 and has effectively failed.
> >
> > XGBoost can still maintain 70%–80%, which shows that the reviewer’s judgement about XGBoost is reasonable. However, compared with various advanced models that are specifically designed for this task, ML methods are indeed more than 20% worse. This is because encrypted traffic is different from other domains: payload encryption leads to randomization and increased entropy, so feature engineering based on information obtained from the payload can be misleading. Our method simultaneously considers both topology and payload, and we also perform dedicated alignment on the plaintext headers within the payload, which mitigates the impact introduced by encryption.

---

> ### Author Response · Authors · 2025-11-22
>
> ### Weakness 5:
> In the experiments, it would be useful to include comparisons with NetMamba, which likewise emphasizes efficiency. In addition, the reported accuracies on ISCX-VPN2016 are noticeably lower than those in the original papers (e.g., YATC and TrafficFormer). This may reflect differences in pretraining data or other factors; please report the exact pretraining datasets used and explain the lower accuracy. Finally, because the performance gap on USTC-TFC2016 is relatively small across methods, adding more datasets would help substantiate the generality of the conclusions.
>
> ### Response to Weakness 5:
> We thank the reviewer for the detailed comments. In response to the suggestions on comparing with NetMamba, explaining the lower accuracy on ISCX-VPN2016, and adding more datasets, we provide the following clarifications and revisions:
>
> (1) **On pretraining and the YATC model**:
> The original YATC publicly released pretrained models have data overlap with the datasets we use. In their paper, they explicitly state that the VPN dataset is divided into two parts: one part of unlabeled data is used for pretraining, and the other part is used for supervised fine-tuning. To avoid potential data leakage and to ensure fair comparison, we re-performed pretraining on CICIDS2017 and a browser traffic dataset, and conducted comparative experiments based on these “non-overlapping” models. The corresponding training hyperparameters, data partitioning schemes, and model weights will later be uniformly uploaded to an anonymous repository to facilitate reproduction and verification.
>
> (2) **On the reasons for the lower accuracy on ISCX-VPN2016**:
> First, YATC uses a 7-class VPN-only ISCX-VPN setting, whereas the mainstream practice (and also the setting adopted in this paper) is a 14-class classification task that includes both VPN and non-VPN, which is inherently more difficult. Second, in its publicly released dataset, YATC adopts a specific sampling strategy that splits the data into a pretraining part and a fine-tuning part, and the actual amount of data used for fine-tuning is significantly smaller than the original raw data. In our experiments, it is difficult to make the data scale and distribution exactly identical, and therefore directly comparing accuracy numbers is not entirely fair.
>
> In addition, TrafficFormer and NetMamba both adopt a relatively aggressive data-augmentation strategy: starting from a single original pcap flow, they generate multiple “different” training samples simply by modifying fields such as the TCP sequence number. Although this can improve accuracy, it in fact changes the effective size and class distribution of the dataset (for example, 500 samples are expanded to 1000), so the comparison results become highly dependent on the specific augmentation strategy. For reasons of fairness and interpretability, when reproducing these methods we do not use this data-augmentation approach; instead, we train strictly based on the original flow splits and counts. We believe that these differences in label space, sampling strategies, and data-augmentation strategies are the main reasons why our absolute accuracy on ISCX-VPN2016 is slightly lower than the values reported in the original YATC and TrafficFormer papers.
>
> (3) **On additional datasets to verify the generality of the conclusions**:
> To alleviate potential concerns arising from the small performance gaps among methods on USTC-TFC2016 alone, we have added experimental results on CipherSpectrum (S&P 2025) in the revised manuscript. This dataset uses traffic with a mixture of three advanced encryption schemes and has already been split by flows, so there is no confounding from different processing choices. As shown in Table 2, S²-ETR still achieves better performance than mainstream methods on this dataset, reaching 98% accuracy. This supports the generalizability of our conclusions from the perspective of newer and more complex traffic distributions.

---

> > ### Author Response · Authors · 2025-11-22
> >
> > ### Question 1:
> > Add more discussion on the motivation for adopting a topology-based approach.
> >
> > ### Response to Question 1:
> > In the revised manuscript, we have expanded the Introduction and Related Work to more clearly explain why we adopt a topology-based approach in encrypted traffic classification. We first summarize the limitations of traditional machine learning methods that rely on handcrafted statistical features, which are hard to tune and generalize poorly to new encryption schemes and application behaviors. We then emphasize that sequence based and image based deep models mainly capture the semantics of individual flow byte sequences, but recent studies [1][2] show that carefully crafted malicious flows can become almost indistinguishable from benign ones after encryption, with highly randomized, high entropy byte sequences that are hard to distinguish at the single flow level. These observations motivate a shift from modeling isolated flows to exploiting relations among flows via topology based analyses. We further point out that, in traffic interaction graphs, most discriminative information lies on edges rather than nodes, and that message passing through intermediate nodes exacerbates over squashing and scalability issues, especially when multiple flows between the same IP pair are collapsed into a single edge embedding. Based on this analysis, we explicitly state that our goal is to jointly model intra flow semantics and inter flow relations, so that topology complements single flow models and leads to more accurate and more generalizable detection across diverse encrypted traffic scenarios (see Section 1 & 2).
> >
> > [1] Zhuoqun Fu, Mingxuan Liu, Yue Qin, Jia Zhang, Yuan Zou, Qilei Yin, Qi Li, and Haixin Duan. Encrypted malware traffic detection via graph-based network analysis. In Proceedings of the 25th International Symposium on Research in Attacks, Intrusions and Defenses, pp. 495–509, 2022.
> > [2] Chuanpu Fu, Qi Li, and Ke Xu. Detecting unknown encrypted malicious traffic in real time via flow interaction graph analysis. In 30th Annual Network and Distributed System Security Symposium, NDSS 2023
> >
> > ### Question 2:
> > Restructure Sections 2 and 3 to present the architecture and key designs more intuitively, avoiding excessive reliance on formulas.
> >
> > ### Response to Question 2:
> > We have restructured Sections 2 and 3 to improve readability and reduce the reliance on mathematical formulas in the main text. Non essential derivations, such as some preliminaries and the full IHL iterative formulas, have been moved to the appendix, while the main text keeps only the key computational steps and necessary notation. Section 2 is now organized as Related Works, highlighting the limitations of existing methods and motivating the combination of topology and semantics, including a description of oversquashing and relevant solutions. Section 3 is reworked around intuitive explanations and figures, with a simple and clear overview subsection added at the beginning of Section 3 to summarize the framework before presenting technical details (see Responses 1 and 3).
> >
> > ### Question 3:
> > Include additional datasets and comparative experiments, such as with traditional machine-learning methods (FlowPrint, AppScanner, raw packet–based XGBoost), and NetMamba.
> >
> > ### Response to Question 3:
> > We have added comparative experiments with traditional machine learning methods, including FlowPrint, AppScanner, and an XGBoost classifier built on raw or simple statistical features, as well as with NetMamba. In addition, we have incorporated more datasets into our evaluation, so that the comparisons cover both weakly encrypted and strongly encrypted traffic. These new results enable a more comprehensive assessment of the proposed method in terms of both accuracy and efficiency, and they are discussed in detail in Section 4.

---

> ### Comment · Reviewer_FCWc · 2025-11-25
>
> Thank you for the detailed revision, it seems to have resolved a large part of the paper's problems, especially the addition of Figure 2, which helps with the readability.
>
> But I think there are still some critical issues in this paper:
>
> 1. The authors claim that previous work faced three main limitations: overfitting to semantic features with poor generalization, weak cross-domain robustness, and constrained scalability to large and complex traffic datasets. However, the experimental section does not seem to provide sufficient proof for 'generalization' or 'cross-domain robustness'. (eg: hypervision uses unsupervised training to prove it doesn't require prior knowledge), the proposed method essentially performs training and testing on each dataset independently. In fact, the results show that several works perform well on their respective datasets and can also support large-scale datasets. I think these 'three main challenges' are poorly extracted. They seem more like challenges faced by certain works and mixed together.
>
> 2. In addition, based on my current understanding of this paper, the core contribution of this work is enriching simple flow interaction graphs with additional information to achieve better classification results. The trade-off, however, is a much heavier graph structure.
>
> Although the authors claim that S2-ETR maintains competitive performance through a hierarchical classifier guided by conditional probabilities, the larger traffic interaction graph appears to be stored in GPU memory, resulting in out-of-parameter overhead. It remains unclear to me how the authors address this specific memory bottleneck.
> I advise the authors to abandon overly complex language and use the most direct and intuitive manner to present their core contributions. There is no need to worry that simplicity makes the paper look unsophisticated; high-quality work should always be like this in fact. As for technical terminology, it can be used to support the core contributions, by indicating which part of the functionality the proposed solution is designed to achieve, or by addressing problems arising from the solution itself.
>
> 3. The authors have added many additional experiments to demonstrate the effectiveness of their method. Although some of the responses differ from my prior experience, the reasoning they provided is logically sound. In the experiments, providing more information indeed leads to a substantial improvement in accuracy. In fact, the ISCXVPN dataset contains some trivial shortcut features, such as port numbers. I assume you likely masked these shortcuts during the comparison, which is reasonable. However, the flow interaction graph that incorporates flow information may have reintroduced these shortcuts, resulting in higher accuracy. As we can observe, the accuracy gap among the S2-ETR variants using 1D-CNN, 2D-CNN, Linear, and ResNet is relatively small. This suggests that the amount of available information has a greater impact than the strength of the model itself. While this effect could be attributed to the proposed architectural design, it remains difficult to rule out the possibility that hidden shortcuts are influencing the results.
>
> 4. The newly added Figure 1 lacks sufficient information, and the usage of the terms 'inter' and 'intra' throughout the paper appears to be problematic.

---

> > ### Author Response · Authors · 2025-11-28
> >
> > We sincerely thank the reviewers for their thoughtful and constructive comments regarding (1) the articulation of our three main challenges, (2) the clarity and practicality of our core contributions and memory efficiency, (3) the concern about potential shortcut features in the flow interaction graph, and (4) the clarity of Figure 1 and terminology usage. We have thoroughly revised the manuscript accordingly and provide detailed point-by-point responses below.
> >
> >
> > ### **1. On the three main challenges**
> >
> > **Response:**
> > We appreciate the reviewer’s insightful comment. Following your suggestion, we have re-examined and refined our problem formulation. We acknowledge that the previously stated “three challenges” were overly broad and could be misinterpreted. We have reorganized our contributions to clearly state the **three precise and technical problems our method directly addresses**:
> >
> > 1. As **traffic encryption and adversarial obfuscation** intensify, **intra-flow** approaches that model individual flow sequences with deep learning exhibit reduced classification accuracy and weakened robustness.
> >
> > 2. **Inter-flow** methods that **incorporate cross IP-pair interactions** and **aggregate contextual signals** can enhance flow representations, but when most salient information is encoded on graph edges, message passing suffers severe over squashing, which constrains model expressivity and harms performance.
> >
> > 3. Methods that jointly leverage **intra-flow semantics and Inter-flow structural dependencies** remain scarce, so designing effective fusion strategies to reconcile per-flow sequential signals with cross-flow structural context is imperative.
> >
> > ---
> >
> > ### **2. On the complexity of the graph structure and the concern about GPU memory bottlenecks**
> >
> > **Response:**
> > We thank the reviewer for pointing this out. In fact, **lightweight graph design and low memory consumption were core considerations** in S$^2$-ETR. We analyze this from both experimental and theoretical perspectives:
> >
> > 1. To make this concrete, we report **full memory profiling** (added to the Appendix):
> >
> > | Stage                 | Python Mem (MB) | GPU Mem (MB)             | Notes                             |
> > | --------------------- | --------------- | ------------------------ | --------------------------------- |
> > | Dataset loading       | 118.04          | —                        | 663,032 flows                     |
> > | Bipartite graph       | 150.40          | 25.29                    | Sparse adjacency = *only* 25.29MB |
> > | Hypergraph build      | 215.75          | —                        | —                                 |
> > | Model init            | 215.75          | 64.89                    | 1.024M parameters                 |
> > | Bipartite propagation | 215.75          | **760.26 (peak 792.39)** | Batch=64                          |
> > | Hypergraph attention  | 215.75          | **761.45 (peak 762.58)** | Batch=64                          |
> >
> > Even with **660k flows**, the sparse adjacency matrix consumes only **25.29MB**. The GPU memory footprint is well within the capacity of mainstream GPUs.
> >
> > Under the same configuration, monitoring of NetMamba shows a GPU peak of 1.25 GB, an average GPU usage of 1.02 GB, CPU memory usage of 732 MB, and 1.87M trainable parameters. Our method requires approximately **38.1%** less peak GPU memory than **NetMamba**.
> >
> > 2. Theoretical perspectives
> > * **Sparse bipartite representation:**
> >   Each flow connects to only two IP nodes. Thus, both storage and message passing are conducted on **sparse matrices**, significantly reducing memory overhead.
> >
> > * **Small number of trainable IP nodes:**
> >   For example, in CICAndMal, there are **663,032 flow nodes** but only **7,914 IP nodes** require learning updates. Thus, each training epoch updates only 7,914 learnable representations, resulting in about 1.024M parameters, which is **45.2% fewer** than the baseline value of 1.87M （**NetMamba**）. This considerably reduces GPU memory requirements compared with dense semantic models.
> >
> > We also simplified overly complex language in the revised paper, following the reviewer’s suggestion.

---

> > > ### Author Response · Authors · 2025-11-28
> > >
> > > ### **3. On the concern that improved accuracy may stem from undiscovered shortcut features**
> > >
> > > **Response:**
> > > Thank you for highlighting this important point.
> > > To eliminate explicit shortcut leakage, **we masked all IPs as 255.255.255.255 and ports as 0** in our experiments.
> > >
> > > To directly test whether the interaction graph reintroduces shortcuts implicitly, we conducted **new experiments** on CipherSpectrum2025, comparing **GCN vs. S2-ETR** under “mask” vs. “no mask” settings.
> > >
> > > | Method     | Mask IP/Port | ACC        | PRE    | REC    | F1     |
> > > | ---------- | ------------ | ---------- | ------ | ------ | ------ |
> > > | **GCN**    | No           | 0.8001     | 0.8174 | 0.8001 | 0.7981 |
> > > |            | Yes          | **0.5549** | 0.7190 | 0.5549 | 0.5696 |
> > > | **S2-ETR** | No           | 0.9725     | 0.9741 | 0.9725 | 0.9723 |
> > > |            | Yes          | **0.9737** | 0.9740 | 0.9737 | 0.9737 |
> > >
> > > **Key findings:**
> > >
> > > The results show that after masking IPs and ports, that is, removing explicit shortcuts, the accuracy of a standard GCN on the traffic interaction graph **drops by 25\%** and becomes only **55.49\%**. This is lower than the machine learning based methods in Table 2 of the main paper (**65.50\%**) and more than **20\% lower** than semantic methods such as CNNs. This indicates that on the traffic interaction graph, after masking IPs and ports, the **influence of shortcuts is already greatly weakened**.
> > >
> > > Furthermore, the results of **S$^2$-ETR** show that when IPs and ports are masked, its performance is slightly **0.1\% higher** than the unmasked version. This is because **CipherSpectrum contains 41025 flows and 778 participating IPs, with 41 classes**. In other words, each IP is very likely to participate in communications of multiple traffic categories. In such a situation, **IP and port information can become noise**, because we consider not only intra flow semantics but also inter flow relationships. The type of one flow associated with an IP may differ from the next. Therefore, the **bipartite design of the traffic interaction graph in S$^2$-ETR, which separates flow nodes, further reduces the impact of shortcuts**.
> > >
> > > ### **4. On the completeness of Figure 1 and the use of “inter” and “intra” terminology**
> > >
> > > **Response:**
> > > We appreciate this feedback. We have:
> > >
> > > * Expanded the **Figure 1 caption** to include full explanations of both inter-flow and intra-flow processing.
> > > * Clarified the usage of “inter-flow” and “intra-flow” in the second paragraph of the introduction.
> > > * Added brief definitions in the caption to prevent ambiguity.
> > >
> > > ---
> > >
> > > ### **Once again, we sincerely thank the reviewer for the detailed comments. They helped us significantly improve both the technical clarity and presentation quality of the paper.**

---

### Official Review · Reviewer_EsDz · 2025-10-31

**Soundness:** 3
**Presentation:** 3
**Contribution:** 3
**Rating:** 6
**Confidence:** 3

**Summary:**

The paper proposes a novel encrypted traffic classification framework named S2-ETR, designed to address core challenges in existing encrypted traffic identification methods. To this end, the authors developed a dual-branch architecture integrating both semantic modeling and topological modeling, while introducing an implicit hypergraph learning mechanism to adaptively model higher-order relationships between nodes.

**Strengths:**

1.The paper presents a clearly defined and practically significant problem motivation, accurately identifies key bottlenecks in current encrypted traffic classification, and proposes a targeted architectural design. The research background and motivations are well-justified and persuasive.
2.The dual-branch architecture integrating semantic and topological branches, along with the introduction of IHL, constitutes the core innovation of this work. This design effectively mitigates the over-squashing issue inherent in traditional GNN models while enhancing the complementary relationship between semantic and topological information.

**Weaknesses:**

1.The statically computed α value fundamentally contradicts the claimed "adaptive learning" capability. Furthermore, while contemporary graph neural networks commonly employ dynamic weighting strategies such as gating mechanisms, this study fails to substantiate the comparative advantages of the fixed-weight approach.
2.The experimental design and argumentation completeness of this paper raise several concerns. Firstly, the comparative experiments on the most challenging large-scale dataset CIC-AndMal2017 are insufficient due to the absence of comparisons with stronger pre-training baseline models like YaTC, which undermines the persuasiveness of the claimed performance superiority. Secondly, the ablation study casts doubt on the effectiveness of the semantic branch, particularly on the USTC-TFC2016 dataset where its removal yields better performance than the full model. Finally, the timeliness of the experimental datasets is inadequate, with most versions being outdated (e.g., from 2016/2017), making it difficult to validate the model's adaptability to contemporary encrypted traffic.

**Questions:**

1.It is recommended to explore at least one dynamic weighting strategy as a comparative baseline to demonstrate the trade-offs against the static method.
2.On large-scale datasets such as CIC-AndMal2017, it is essential to include comparisons with widely recognized strong baseline models (e.g., YaTC). This is a critical prerequisite for validating whether the method's performance achieves state-of-the-art levels.
3.Regarding the anomalous observation in the ablation study where "removing the semantic branch resulted in improved performance," further tests should be conducted on additional datasets to determine whether this is an isolated case or a general phenomenon.
4.To address concerns about the model's capability to handle modern encrypted traffic, it is strongly recommended to incorporate validation experiments on recently released datasets.

---

> ### Author Response · Authors · 2025-11-22
>
> ### Weakness 1:
> The statically computed α value fundamentally contradicts the claimed "adaptive learning" capability. Furthermore, while contemporary graph neural networks commonly employ dynamic weighting strategies such as gating mechanisms, this study fails to substantiate the comparative advantages of the fixed-weight approach.
>
> Response to weakness 1:
>
> We thank the reviewer for highlighting this important issue. In the original submission, we used a statically computed α to quantify the sparsity of the IP-Flow graph, which is then used to balance the feature fusion between the structural and semantic branches. The “adaptive learning” capability of our method mainly lies in the fact that the semantic hypergraph and the bipartite-graph topology are iteratively updated from the raw data and automatically adjusted to an optimal configuration, without requiring additional data constraints or supervision. As the reviewer correctly pointed out, our previous wording was not sufficiently precise; we have revised the manuscript accordingly, and the modifications are marked in blue in the updated version.
>
> In addition, we introduce two baseline fusion mechanisms, namely a gating mechanism and the AFF mechanism, which are briefly described in Section 3 and evaluated experimentally. These experiments compare static and dynamic fusion weights in terms of both performance and inference time. As shown in Table 12, our method achieves 97% accuracy, which is 3.5% higher than both the gating and AFF baselines, while also improving inference speed by 70%. We attribute this to the fact that the learnable fusion weights in the dynamic variants tend to converge prematurely and become trapped in suboptimal local minima, whereas the static weight, which is designed to reflect the global topological sparsity, yields more stable and consistently superior performance.
>
> ### Weakness 2:
>
> Response to weakness 2:
> We appreciate the reviewer’s constructive comments regarding the experimental design and completeness. In response, we have conducted additional experiments and clarified the corresponding analyses as follows:
>
> (1) **Comparison with stronger pre-training baselines on CIC-AndMal2017.**
> We agree that including a stronger pre-training baseline is important for supporting the claimed performance advantages on the most challenging large-scale dataset CIC-AndMal2017. Since NetMamba adopts a preprocessing pipeline similar to YaTC while representing a more recent and efficiency-oriented pre-training approach, we believe that using NetMamba as a baseline on large-scale data is more informative and persuasive. We have therefore added experiments with NetMamba on CIC-AndMal2017 to compare our method with a state-of-the-art pre-training model. The results show that NetMamba achieves 43.14% accuracy, with ACC being 0.4% lower than our 1D-Adapter, while its Recall is higher by 1%. However, NetMamba requires a large external corpus as prior knowledge. In contrast, our method achieves performance comparable to such advanced pre-training models **without** relying on any additional external data, which we believe highlights the advantages of our architectural design and model structure.
>
> (2) **Effectiveness of the semantic branch on USTC-TFC2016.**
> We acknowledge the reviewer’s concern that, on the USTC-TFC2016 dataset, removing the semantic branch yields slightly better ACC than the full model, which may appear to challenge the effectiveness of the semantic branch. This phenomenon is in fact a dataset-specific artifact. USTC-TFC2016 has a highly skewed distribution: each IP almost exclusively sends or receives only a small number of flows, and in practice each IP is associated with (nearly) a single type of flow. As shown in the results table 2, due to this structural simplicity, almost all baselines already achieve over 90% accuracy, with about half exceeding 95%. In such an extreme setting, the model can obtain strong performance using only the structural (IP-level) connectivity, and adding a semantic branch may introduce additional noise instead of useful complementary information. Nonetheless, our full model suffers less than a 0.4% drop in ACC compared with the variant without the semantic branch, and overall remains highly competitive. More importantly, such degenerate cases—where each IP is tied to essentially one traffic type—are very rare in real-world encrypted traffic scenarios, where semantic diversity is much higher and the semantic branch becomes more beneficial.

---

> > ### Author Response · Authors · 2025-11-22
> >
> > ### Response to Weakness 2 (3)
> >
> > (3) **Timeliness and realism of the experimental datasets.**
> > We used VPN2016 and USTC-TFC2016 primarily to enable fair comparison with the most widely adopted baselines in the literature. However, we fully agree that these datasets are relatively outdated (e.g., from 2016/2017) and alone are not sufficient to demonstrate adaptability to contemporary encrypted traffic. To address this, we have updated our experimental protocol by introducing a new dataset, **CipherSpectrum2025**, which incorporates three mixed advanced encryption schemes to better emulate modern encrypted traffic patterns. The corresponding experiments and results have been added to Section 4.2 and highlighted in blue in the revised manuscript. Notably, S$^2$-ETR (ResNet) pushes ACC/F1 to 0.9802/0.9802 on CipherSpectrum2025, outperforming all baseline models. This further narrows the error margin against already strong baselines and confirms the effectiveness and robustness of S$^2$-ETR in more realistic, up-to-date encrypted traffic scenarios.
> >
> > ### Question 1:
> > It is recommended to explore at least one dynamic weighting strategy as a comparative baseline to demonstrate the trade-offs against the static method.
> >
> > Response to question 1:
> > We appreciate the reviewer’s suggestion and have accordingly included two representative dynamic fusion baselines in our revised manuscript: a gating mechanism and the AFF (Attentional Feature Fusion) mechanism. Both methods are briefly introduced in Section 3, and we conduct a systematic empirical comparison between these dynamic weighting strategies and our static weighting scheme in terms of both performance and computational overhead. As shown in Table 12, our static fusion strategy achieves 97% accuracy, outperforming the gating and AFF baselines by 3.5 percentage points, while simultaneously improving inference speed by approximately 70%. This is because the learnable fusion weights in the dynamic variants tend to converge prematurely and become trapped in suboptimal local minima, whereas the static fusion weight, which is designed to reflect the global topological sparsity of the IP-Flow graph, maintains more stable and consistently superior performance. These results explicitly demonstrate the trade-off between dynamic flexibility and stability/efficiency, and support our choice of a static weighting strategy.
> >
> > ### Question 2:
> > On large-scale datasets such as CIC-AndMal2017, it is essential to include comparisons with widely recognized strong baseline models (e.g., YaTC). This is a critical prerequisite for validating whether the method's performance achieves state-of-the-art levels.
> >
> > Response to question 2:
> > We fully agree that comparisons with strong pre-training baselines on CIC-AndMal2017 are crucial for validating the competitiveness of our method. In the revised manuscript, we therefore introduce **NetMamba** as a representative strong pre-training baseline. NetMamba adopts a preprocessing pipeline similar to YaTC while being a more recent and efficiency-oriented pre-training approach, which makes it a suitable and persuasive benchmark on large-scale data. We conduct experiments with NetMamba on CIC-AndMal2017 and compare it directly with our method. The results show that NetMamba achieves 43.14% accuracy, with ACC being 0.4 percentage points lower than our 1D-Adapter, while its Recall is higher by about 1 percentage point. However, NetMamba requires a large external corpus as prior knowledge, whereas our approach achieves comparable performance **without** relying on any additional external data. This observation highlights the architectural advantages and structural efficiency of our method and supports the claim that it reaches state-of-the-art–level performance under a more constrained and practical setting.

---

> > > ### Author Response · Authors · 2025-11-22
> > >
> > > ### Question 3:
> > > Regarding the anomalous observation in the ablation study where “removing the semantic branch resulted in improved performance,” further tests should be conducted on additional datasets to determine whether this is an isolated case or a general phenomenon.
> > >
> > > Response to question 3:
> > > We appreciate the reviewer’s careful reading of the ablation results. The observation that removing the semantic branch slightly improves accuracy on **USTC-TFC2016** is indeed a special case caused by the extreme simplicity of this dataset’s topology. In USTC-TFC2016, each IP address typically sends or receives only a very small number of flows and is almost always associated with a single traffic type. Under such conditions, the structural connectivity alone already provides nearly complete discriminative information. As a result, the model can achieve strong performance purely from the IP-level structure, while the additional semantic branch may introduce mild noise rather than complementary information. Even so, the full model on USTC-TFC2016 suffers less than a 0.4% drop in accuracy compared with the variant without the semantic branch, and overall performance remains high, with most baselines exceeding 90% and about half exceeding 95%.
> > >
> > > To further verify that this behavior is not a general phenomenon, we performed two additional sets of experiments:
> > >
> > > * **Adapter capacity analysis.** In the main experiments shown in Table.2, our full model uses a 2D-CNN as the Adapter. When we replace this with a deeper and more expressive ResNet-based Adapter, the accuracy on USTC-TFC2016 improves to **98.12%**, indicating that a more suitable adapter can effectively suppress the noise and better leverage the semantic information.
> > > * **Cross-dataset validation.** We extended the ablation study to more realistic and diverse benchmarks, including **CICIoT2023** and **ISCXVPN2016**, as shown in Table 10. On both datasets, adding the semantic branch consistently improves performance by **more than 2 percentage points** (in ACC and F1), demonstrating that the semantic branch is beneficial in typical encrypted traffic scenarios.
> > >
> > > Taken together, these results confirm that the USTC-TFC2016 case is an artifact of its overly simple and degenerate topology rather than a general weakness of our design, and that the semantic branch plays an important and effective role on more representative datasets.
> > >
> > > ### Question 4:
> > > To address concerns about the model's capability to handle modern encrypted traffic, it is strongly recommended to incorporate validation experiments on recently released datasets.
> > >
> > > Response to question 4:
> > > We thank the reviewer for this important suggestion. In the revised version, we have expanded our experimental protocol to include **recent and more realistic encrypted traffic datasets**. Specifically, in addition to CICIoT2023, we incorporate the newly constructed **CipherSpectrum2025** benchmark, which combines three advanced encryption schemes to more faithfully reflect contemporary encrypted traffic patterns. The corresponding experimental setup and results have been added to Section 4.2 and highlighted in blue in the manuscript.
> > >
> > > On CipherSpectrum2025, our S$^2$-ETR (ResNet) variant achieves ACC/F1 of **0.9802/0.9802**, outperforming all baseline models and further narrowing the performance gap relative to already strong competitors. Together with the results on CICIoT2023, these experiments demonstrate that our method maintains high accuracy and robustness on up-to-date encrypted traffic, thereby addressing the concern about timeliness and validating the model’s adaptability to modern deployment scenarios.

---

### Official Review · Reviewer_uxP3 · 2025-10-31

**Soundness:** 2
**Presentation:** 2
**Contribution:** 2
**Rating:** 4
**Confidence:** 4

**Summary:**

This paper propose S2-ETR , a framework that integrates traffic semantics with communication topology graph for encrypted traffic classification (ETC), effectively alleviating the over-squashing and long-range dependence problems in traditional graph neural networks. Experimental results show that S2-ETR achieves SOTA performance.

**Strengths:**

+ It provides mathematical proof that the HBG structure is better than traditional IP topology in terms of information dissemination efficiency, which enhances the persuasion of the paper.
+ It was compared with 11 baselines on multiple public datasets (such as CIC-IoT2023, ISCX-VPN2016, USTC-TFC2016), covering different topologies and data sizes.

**Weaknesses:**

+ The expression of HBG and IHL  is very dense, and some formulas and illustrations are not intuitive enough. Some terms (such as "implicit hypergraph" and "semantic semantic hyperedges") lack clear definitions.
+ The datasets used are old (year 2016) and do not represent the realistic characteristics of real-world encrypted traffic. Please consider evaluating it on the latest dataset, such as CipherSpectrum[S&P 2025].
+ The proposed method alleviates the over-squashing problem, but lacks discussion and comparison with other over-squashing solutions.

**Questions:**

+ What does semantics mean in encrypted traffic, Why HBG can capture semantics, and how to prove that semantics are indeed captured.
+ Over-squashing is a classic topic in graph neural networks, but the paper lacks discussion and comparison of common solutions. Please clearly state the contribution and innovation of the method in solving over-squashing problems.
+ Why select 500 flows per class? Is this can reflect the distribution of the real world?
+ Why consider CNN, Linear and ResNet? Without considering other network structures, such as transformer.
+ The best recall for the ISCX-VPN2016 dataset in Table 3 should be S2-ETR (ResNet) rather than S2-ETR (2D-CNN).
+ Why S2-ETR (ResNet) performs worse than S2-ETR (2D-CNN) on CIC-IoT2023.
+ Implementation details are inconsistent with settings disclosed in the code, such as epoch, batch size for each datasets.

---

> ### Author Response · Authors · 2025-11-22
>
> ### Weakness 1:
> The presentation of HBG and IHL is rather dense, and some formulas and figures are not sufficiently intuitive. Several terms (e.g., “implicit hypergraph” and “semantic hyperedges”) lack clear definitions.
>
> Response to Weakness 1.
> We thank the reviewer for pointing out the issues in our presentation. In the revised manuscript, we have made the following changes (highlighted in blue in the revision PDF):
>
> 1. **Reorganized and simplified key formulas for HBG and IHL.**
>    We have rearranged and simplified the core equations of HBG and IHL to avoid introducing too many symbols in a single line. Part of the derivation has been moved to the appendix, and only the essential expressions are kept in the main text, thereby reducing the cognitive load for readers.
>
> 2. **Improved explanation of the framework figure.**
>    In Section 3.1, we now provide a step-by-step explanation of the framework diagram, clarifying the overall pipeline and the underlying principles of our method.
>
> 3. **Added clearer terminology definitions.**
>
>    (1) We define **“implicit hypergraph”** as the collection of hyperedges that is implicitly induced in the feature space by inter-flow similarity/attention weights [1][2], rather than a hypergraph explicitly constructed from hand-crafted prior rules (Appendix B.1).
>    (2) We clarify that **semantics** refer to features that reflect network behavior patterns, which are implicitly encoded in observable byte streams, packet sequences, and flow statistics (Section 2, Related Work).
>    (3) We define **semantic hyperedges** as hyperedges that connect multiple flows sharing similar high-order behavioral semantics—for example, flows belonging to the same application/service or exhibiting similar temporal patterns.
>
> We believe these modifications substantially improve the readability and intuitiveness of the HBG and IHL components.
>
> [1] *Machine Learning-Powered Encrypted Network Traffic Analysis: A Comprehensive Survey*, IEEE Communications Surveys & Tutorials 2023.
> [2]*Rosetta: Enabling Robust TLS Encrypted Traffic Classification in Diverse Network Environments with TCP-Aware Traffic Augmentation,* USENIX Security '23.
>
>
> ### Weakness 2:
> The datasets used are old (year 2016) and do not represent the realistic characteristics of real-world encrypted traffic. Please consider evaluating it on the latest dataset, such as CipherSpectrum[S&P 2025].
>
> Response to Weakness 2.
>
> We thank the reviewer for the valuable suggestion regarding our dataset choices.
>
> 1. **On using ISCX-VPN2016 and other “older” datasets.**
>     We initially chose ISCX-VPN2016 and other classic benchmarks because they are widely used in the encrypted traffic classification literature, which allows for fair and direct comparison with a large number of existing methods. In addition, these datasets contain diverse VPN/non-VPN traffic and multiple application scenarios, making them suitable for evaluating the generalization ability of our model across different categories of encrypted traffic from an algorithmic perspective.
>
> 2. **Evaluation on newer datasets (including CipherSpectrum).**
>     We fully agree with the reviewer that the nature of encrypted traffic has evolved rapidly in recent years, and emerging protocols and applications can significantly change traffic statistics. Motivated by this, we have added experiments on CipherSpectrum2025 and CICIoT2023, and reported the results in Section 4.2 of the revised manuscript. CipherSpectrum adopts three advanced encryption schemes and exhibits lower IP/(flow+IP) values and a denser topology, thus better reflecting modern encrypted traffic.
>
>     The experimental results show that our method achieves up to 98.02% ACC on CipherSpectrum2025, outperforming strong baselines such as YaTC and TrafficFormer by 1%–2%, and significantly surpassing traditional ML-based approaches. These results demonstrate that our model maintains state-of-the-art performance and strong robustness even on recent, more realistic encrypted-traffic datasets.

---

> > ### Author Response · Authors · 2025-11-22
> >
> > ### Weakness 3:
> > The proposed method alleviates the over-squashing problem, but lacks discussion and comparison with other over-squashing solutions.
> >
> > Response to weakness 3:
> > We thank the reviewer for raising this important point. In the revised manuscript, we have strengthened both the conceptual discussion and empirical comparison on over-squashing from two aspects:
> >
> > 1. **Additional related work and theoretical discussion.**
> >    In Section 2 (Related Work), we now explicitly discuss both *classical* and *recent* approaches to mitigating over-squashing, including curvature-based methods such as SRDF [3] and community/feature-similarity–guided rewiring such as ComFy [4]. Based on this, we clarify that our method takes a complementary route: by introducing an **implicit hypergraph** and **semantic hyperedges**, we increase *higher-order connections among multiple nodes* at the structural level, which is effectively equivalent to shortening the path length between distant nodes in the graph, thereby alleviating over-squashing. A more detailed theoretical explanation is provided in Appendix B.2.
> >
> > 2. **New comparative experiments.**
> >    In Section 4.4 of the revised manuscript, we add a direct comparison with SRDF [3] and ComFy [4]. Specifically, we apply their rewiring strategies and our HBG/IHL modules on the **same backbone** and analyze the performance under identical settings, as reported in Table 11. Our method achieves **97.8%** accuracy, which is **significantly higher** than ComFy (94.70%) and **slightly better** than SRDF (97.51%), demonstrating a strong capability to mitigate over-squashing while maintaining superior task performance.
> >
> > [3] *Understanding over-squashing and bottlenecks on graphs via curvature*, ICLR 2022.
> > [4] *GNNs Getting ComFy: Community and Feature Similarity Guided Rewiring*, ICLR 2025.
> >
> > ### Question 1:
> > What does semantics mean in encrypted traffic, Why HBG can capture semantics, and how to prove that semantics are indeed captured.
> >
> > Response to Question 1:
> > We thank the reviewer for raising this point and clarify how we define and validate semantics in our setting as follows:
> >
> > 1. **Definition of semantics in encrypted traffic.**
> > In our work, the semantics of encrypted traffic refer to features that reflect network behavior patterns, which are implicitly encoded in observable byte streams, packet sequences, and flow statistics [1] (Section 2, Related Work). For example, at the protocol level, TCP semantics correspond to patterns such as the negotiation of specific TCP options, congestion-control interactions, and packet retransmissions [2], which are known to remain robust and invariant across diverse network environments.
> >
> > 2. **Why HBG can capture semantics.**
> > In our model, HBG captures semantics by grouping behaviorally similar flows into shared hyperedges. The implicit hypergraph is constructed from attention scores over high-dimensional flow features, so flows with similar protocol- and behavior-level patterns are more likely to be connected by the same hyperedge. During hyperedge-based message passing (high-order neighborhood aggregation), each flow representation is updated using information from these groups of similar flows, rather than from isolated pairwise neighbors. This process drives the learned embeddings toward application/behavior-pattern–level abstractions that are more consistent within the same semantic class and more distinguishable across different classes.
> >
> > 3. **Empirical evidence that HBG/IHL capture semantics.**
> > As shown in the ablation study in Appendix G.1 (Table 10), we compare (a) a variant without HBG/IHL, (b) a variant using only a simple kNN graph, and (c) the full model with semantic hyperedges. Without IHL-based semantic modeling, performance drops by about 40%, whereas introducing semantic hyperedges raises the accuracy to 86.50%, indicating that IHL effectively learns semantic representations of traffic. In addition, t-SNE/UMAP visualizations of flow embeddings (with vs. without HBG/IHL, Fig. 6) show that, after applying HBG/IHL, flows exhibit clearer clustering by application/behavior, further supporting the effectiveness of our semantic hypergraph modeling.

---

> > > ### Author Response · Authors · 2025-11-22
> > >
> > > ### Question 2:
> > > Over-squashing is a classic topic in graph neural networks, but the paper lacks discussion and comparison of common solutions. Please clearly state the contribution and innovation of the method in solving over-squashing problems.
> > >
> > > Response to Question 2:
> > > This concern is closely related to “Weakness 3”. In the revised manuscript, we have provided a unified clarification (highlighted in blue): we explicitly discuss the over-squashing issue in encrypted-traffic interaction graphs in Section 1 (Introduction); we add a subsection in Section 2 (Related Work) on “the relationship between our method and existing over-squashing solutions”; in Section 3.4, we clearly state that our HBG + IHL can be viewed as a generic structural module for mitigating over-squashing, by introducing high-order connections and hierarchical aggregation so that semantically related distant nodes can exchange information within shorter effective paths; and in Section 4 (Experiments), we additionally compare our method with two representative over-squashing mitigation baselines, SRDF and ComFy.
> > >
> > > ### Question3:
> > > Why select 500 flows per class? Is this can reflect the distribution of the real world?
> > >
> > > We thank the reviewer for pointing out this potential issue. Our considerations and corresponding revisions are as follows:
> > >
> > > Rationale for sampling 500 flows per class.
> > > This setting is mainly adopted for CICIoT2023 and USTC-TFC2016, where the class distribution is highly imbalanced. If we directly use the full datasets, the majority classes would dominate the training process, making it difficult to fairly compare the intrinsic merits of different methods. Therefore, we choose a moderate downsampling threshold (500 flows per class) to alleviate extreme imbalance. Meanwhile, using around 500 samples per class is a common practice in encrypted traffic classification, providing a relatively balanced setting in which different methods can be fairly compared in terms of their ability to distinguish between various types of encrypted traffic.
> > >
> > > Relation to real-world distributions.
> > > We acknowledge that such “uniform per-class sampling” cannot fully reflect the true distribution of real-world traffic. To address the reviewer’s concern, we have added full-data experiments on CipherSpectrum and CICIoT2023, where models are trained directly on the original imbalanced distributions.
> > >
> > > In particular, to systematically study the impact of flow sampling on performance, we further analyze results under different traffic scales on CICIoT2023. We construct a series of settings with different sampling ratios, and the results are reported in Table 6. We additionally evaluate 2–10 different scales, covering a wide range of distributions from very sparse to relatively dense. The experimental results show that across a broad range of flow counts and distribution densities, the proposed method remains robust, indicating that it can handle sparse and highly imbalanced traffic distributions commonly encountered in real-world scenarios.

---

> > > > ### Author Response · Authors · 2025-11-22
> > > >
> > > > ### Question4:
> > > > Why consider CNN, Linear and ResNet? Without considering other network structures, such as transformer.
> > > >
> > > > Response:
> > > > We thank the reviewer for this insightful comment. In fact, our HBG/IHL module is model-agnostic: any architecture that can process 2D sequential features can in principle serve as an adapter to extract semantic information.
> > > >
> > > > (1) Reason for the current backbone choices.
> > > > Our main goal is to demonstrate the generality of HBG/IHL, so we deliberately choose four representative backbones with different capacities and inductive biases:
> > > >
> > > > (a) Linear: the simplest linear classifier, used as a weak baseline to observe the upper bound of gains brought purely by structural enhancement.
> > > >
> > > > (b) 1D-CNN: a widely used architecture for temporal feature modeling, representing sequence feature extraction.
> > > >
> > > > (c) 2D-CNN: a standard choice in traffic-image / spatio-temporal modeling, representing local convolutional feature extraction.
> > > >
> > > > (d) ResNet: a deeper convolutional network with residual connections, representing stronger, deeper CNN architectures.
> > > >
> > > > By inserting the same HBG/IHL module into these four heterogeneous backbones, we aim to show that the proposed module is largely independent of, and transferable across, different base architectures.
> > > >
> > > > (2) Regarding Transformer-based architectures.
> > > > We fully agree that Transformers are well suited to modeling long-range dependencies, and that several works have applied them to encrypted traffic sequence modeling. In this work, however, our focus is on validating the structural contribution of HBG/IHL across standard CNN-type baselines that are already widely adopted and have mature, comparable implementations on our datasets. Incorporating Transformers would introduce additional architectural and hyperparameter complexity, making it harder to isolate the effect of HBG/IHL itself.
> > > >
> > > > Furthermore, in Section 3.1 we explicitly state that any architecture capable of processing two-dimensional sequences can serve as the adapter, and we justify our choice of these four backbones so that the effect of HBG/IHL can be evaluated in a controlled and comparable manner.
> > > >
> > > > ### Question5:
> > > > The best recall for the ISCX-VPN2016 dataset in Table 3 should be S2-ETR (ResNet) rather than S2-ETR (2D-CNN).
> > > >
> > > > Response:
> > > > We thank the reviewer for carefully checking the table. After re-examining our experimental logs and the table formatting, we confirm that there is indeed a labeling/typographical error (or an incorrect value reference) in Table 3, which led to the wrong model being marked as having the best recall on ISCX-VPN2016. In the revised manuscript, we have corrected the corresponding value and/or highlight to ensure that Table 3 is consistent with the actual experimental results. We have also systematically re-checked all tables to avoid similar issues in the future.
> > > >
> > > > We sincerely apologize for any confusion this may have caused.
> > > > ### Question 6:
> > > > Why S2-ETR (ResNet) performs worse than S2-ETR (2D-CNN) on CIC-IoT2023.
> > > >
> > > > Response:
> > > > We thank the reviewer for this question.
> > > >
> > > > On CIC-IoT2023, some subtasks exhibit relatively simple feature patterns and/or limited training samples. In such settings, the higher capacity of a deep ResNet backbone is more prone to overfitting, whereas a simpler 2D-CNN backbone can generalize better. Consistently, a vanilla 2D-CNN trained directly on CIC-IoT2023 already achieves 67.25% accuracy, which even surpasses ET-BERT and TrafficFormer, which are more complex byte-sequence–based models. Together with the class imbalance and label noise present in CIC-IoT2023, these factors tend to amplify overfitting in more complex architectures, which explains why S2-ETR (ResNet) underperforms S2-ETR (2D-CNN) on this dataset.
> > > > ### Question 7:
> > > > Implementation details are inconsistent with settings disclosed in the code, such as epoch, batch size for each datasets.
> > > >
> > > > Response:
> > > > We thank the reviewer for carefully examining our code and pointing out this issue.
> > > >
> > > > After a thorough check, we confirm that there are indeed minor inconsistencies between the implementation details described in the paper and some configuration files in the repository. The main reason is that, in the early stage of our experiments, we tried several settings of training epochs and batch sizes for different datasets. The final results reported in the paper were obtained with 120 epochs, patience = 20, and batch size = 32, while some of the experimental scripts still retained hyperparameter settings from earlier tuning runs. The actual experiments for the paper were conducted using a separate configuration file, which may cause confusion when reading the code. In the revised version, we systematically unify and document the final experimental settings for all datasets and models in the paper.
> > > >
> > > > We sincerely apologize for the confusion this may have caused and greatly appreciate the reviewer’s help in improving the reproducibility and rigor of our work.

---

### Meta-Review · Area_Chair_sTTk · 2026-01-02

**Summary:**

The reviews are predominantly negative, specifically from 4, 3, 2, and 6, with the majority expressing dissatisfaction in Weakness parts. Consequently, uxP3 has expressed concerns about the old dataset and lack of comparison with other over-squashing solutions.
EsDz and FCWc continue to raise concerns regarding the experimental design and argumentation completeness. FCWc further shows concerns about motivation for adopting a topology-based approach. T3wc express the concerns of theoretical analysis. All reviewers share common concerns about limted comparison. Although the authors have tried the best to answer such concerns, only part of these substantial concerns remain addressed—and likely cannot be adequately addressed—in the submitted rebuttal. Therefore, I persist in recommending the rejection of this paper in its current form.

**Reviewer Concerns:**

Most concerns of  Reviewer uxP3 and EsDz have been solved in the rebuttal, since the authors have provided a detailed explanation and experiment comparison in the rebuttal. However, for FCWc and T3wc, only part of concerns have been well sovled.

**Reviewer Scores:**

uxP3 may increase the score to 6.
EsDz may keep the original score of 6.
 FCWc may increase the score to 3.
T3wc may keep the original score of 4.

---

### Decision · Program_Chairs · 2026-01-26

Reject